# Professor X: Manipulating EEG BCI with Invisible and Robust Backdoor Attack

## Abstract

While electroencephalogram (EEG) based brain-computer interface (BCI) has been widely used for medical diagnosis, health care, and device control, the safety of EEG BCI has long been neglected. In this paper, we propose **Professor X**, an invisible and robust "mind-controller" that can arbitrarily manipulate the outputs of EEG BCI through backdoor attack, to alert the EEG community of the potential hazard. However, existing EEG attacks mainly focus on single-target class attacks, and they either require engaging the training stage of the target BCI, or fail to maintain high stealthiness. Addressing these limitations, Professor X exploits a three-stage clean label poisoning attack: **1)** selecting one trigger for each class; **2)** learning optimal injecting EEG electrodes and frequencies strategy with reinforcement learning for each trigger; **3)** generating poisoned samples by injecting the corresponding trigger's frequencies into poisoned data for each class by linearly interpolating the spectral amplitude of both data according to previously learned strategies. Experiments on datasets of three common EEG tasks demonstrate the effectiveness and robustness of Professor X, which also easily bypasses existing backdoor defenses. Code will be released soon.

## 1 Introduction

Electroencephalogram (EEG) is a neuroimaging technology to record of the spontaneous electrical activity of the brain. EEG-based brain-computer interface (BCI) has been widely used in medical diagnosis (Ahmad et al., 2022), healthcare (Jafari et al., 2023), and device control (Lorach et al., 2023; Altaheri et al., 2023). While most EEG community researchers devote themselves to advancing the performance of EEG BCI, the safety of EEG BCI has long been neglected. Inspired by Professor X[1], a superhuman with the ability to control other's minds, we wonder whether a malicious adversary can arbitrarily manipulate the outputs of EEG BCI like him. It will be severely dangerous if so. Backdoor attack (BA), where an adversary injects a backdoor into a model to control its outputs for inference samples with a particular trigger, offers a feasible approach (Doan et al., 2022).

However, designing an effect and stealthy BA for EEG modality is not trivial for three difficulties, resulting in three questions. **D1**: Low signal-to-noise ratio (SNR) and heterogeneity in EEG format (*i.e.*, the montage and sampling rate of EEG recordings) are major obstacles. **Q1**: How to develop a generalizable BA for various EEG tasks (usually have different EEG formats)? **D2**: Previous studies demonstrated for different EEG tasks, different critical EEG electrodes and frequencies strongly related to the performance of EEG BCI (Parvez & Paul, 2014; Jana & Mukherjee, 2021; Baig et al., 2020; Herman et al., 2008), indicating that the trigger-injection strategy (*i.e.*, which electrodes and frequencies to inject triggers) inevitably affects the performance of BA. **Q2**: How to find the optimal strategy for different EEG tasks? **D3**: Certain classes of EEG have specific morphology that can easily be identified by human experts, *e.g.*, in epilepsy detection, the EEG during the ictal phase contains more spike/sharp waves than those during the normal state phase (Blume et al., 1984). **Q3**: How to maintain the consistency of the label and the morphology?

The first BA for EEG modality is demonstrated in Fig 1 (a), where the narrow period pulse (NPP) signals are added as the trigger for single-target class attacks (Meng et al., 2023; Jiang et al., 2023b). To generate invisible triggers, the adversarial loss is applied to learn a spatial filter as the trigger

---

[1]https://en.wikipedia.org/wiki/Professor_X

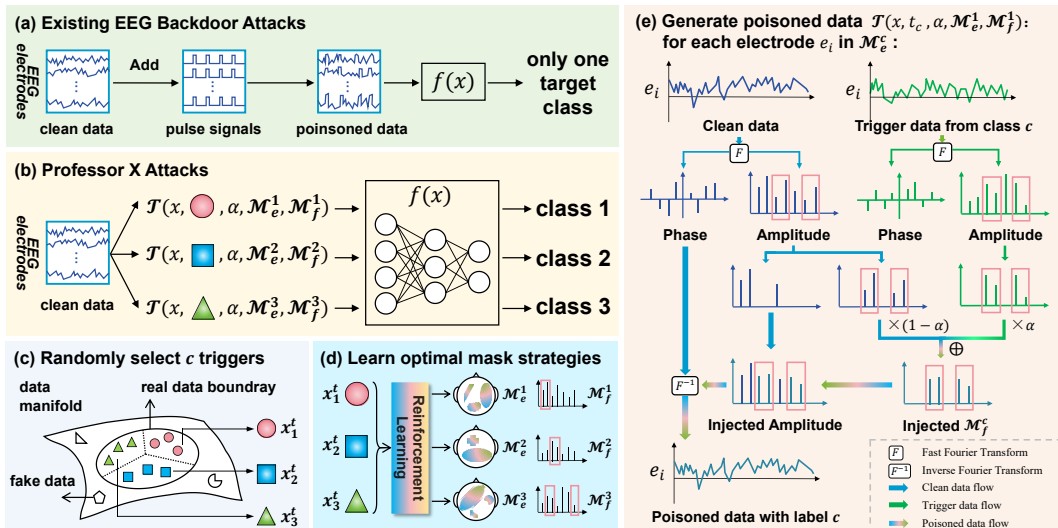

Figure 1: (a) The payloads of the existing backdoor attacks. (b) The payloads of Professor X, which can arbitrarily manipulate the outputs of EEG BCI models.(c)-(e) The framework of Professor X: (c) The trigger selection and EEG data distribution from the view of manifold learning. (d) Learning optimal electrodes and frequencies injection strategies. (e) The generation process.

function (Meng et al., 2024). Recently, some BA for time series (EEG signal is a kind of time series) adopt generative adversarial net (GAN) to produce poisoned data (Ding et al., 2022; Jiang et al., 2023c). However, there is rich information in the frequency domain of EEG (Arroyo & Uematsu, 1992; Kostyunina & Kulikov, 1996; Salinsky et al., 1991; Muthukumaraswamy, 2013). No matter whether these BA are stealthy or not, they all inject unnatural perturbation in the temporal domain, which will inevitably bring unnatural frequency into the real EEG frequency domain.

In this paper, we propose a novel backdoor attack framework **Professor X** to address **Q1**, which injects triggers in the frequency domain and is generalizable to various EEG tasks. Specifically, Professor X is a three-stage clean label poisoning attack demonstrated in Fig 1 (c-e): **1)**: selecting $c$ triggers from $c$ classes. Since these triggers is all real EEG, their frequency are all real, the poisoned EEG (injected with triggers' frequency) is real, as shown in Fig 2(b). **2)**: learning optimal injecting strategy for each trigger with reinforcement learning to enhance the performance of EEG BA, addressing **Q2**. **3)**: generating poisoned data by injecting each trigger's frequency into clean data whose class is the same as the trigger's class, which does not introduce any unreal frequency from other EEG types and maintains the consistency of the label and morphology, addressing **Q3**.

The main contributions of this paper are summarized below:

- We propose a novel backdoor attack for EEG BCI called **Professor X**, which can attack arbitrary class while preserving stealthiness without engaging the training stage .
- To the best of our knowledge, it is the first work that considers the efficacy of different EEG electrodes and frequencies in EEG backdoor attacks.
- Extensive experiments on three EEG BCI datasets demonstrate the effectiveness of Professor X and the robustness against several common preprocessing and backdoor defenses.

## 2 RELATED WORK

### 2.1 BACKDOOR ATTACKS

Backdoor attacks has been deeply investigated in image processing filed (Weber et al., 2023; Yu et al., 2023; Yuan et al., 2023). BadNets (Gu et al., 2019) is the first BA, where the adversary maliciously control models to misclassify the input images contain suspicious patches to a target class. Other non-stealthy attacks include blended (Chen et al., 2017) and sinusoidal strips based (Barni et al.,

2019). To achieve higher stealthiness, some data poisoning BA were developed, including shifting color spaces (Jiang et al., 2023a), warping (Nguyen & Tran, 2020b), regularization (Li et al., 2020) and frequency-based (Zeng et al., 2021; Wang et al., 2022; Hammoud & Ghanem, 2021; Hou et al., 2023; Feng et al., 2022; Gao et al., 2024). Other stealthy attacks (Nguyen & Tran, 2020a; Doan et al., 2021) generate invisible trigger patterns by adversarial loss, which requires the control of the model's training process. To attack multi-target class with high stealthiness, Marksman backdoor (Doan et al., 2022) generates sample-specific triggers by co-training target model and trigger generation model, needing fully control of the training stage. Moreover, generating trigger patterns with a neural network for each sample is time-consuming and unable to use in real-time systems.

## 2.2 BACKDOOR ATTACKS FOR EEG BCI

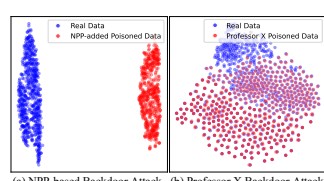

Figure 2: t-SNE visualization.

Recently, the EEG-based BCIs have shown to be vulnerable to BA (Meng et al., 2023; Jiang et al., 2023b; Meng et al., 2024). The NPP signals are added to clean EEG to generate non-stealthy poisoned samples in (Meng et al., 2023; Jiang et al., 2023b), which significantly modifies the spectral distribution (as shown in Fig 2 (a)) and results in low stealthiness. From the view of manifold learning in Fig 1 (a), NPP-added EEG are fake data. To generate more stealthy poisoned data which stay in the real data boundary. The adversarial loss has been applied backdoor EEG BCI (Meng et al., 2024) and time series (Ding et al., 2022; Jiang et al., 2023c). BackTime proposed to generate trigger patterns for each input data using a bi-level optimization (Lin et al., 2024). But these methods can only attack a single target class and require controlling the training process of the backdoor/surrogate models, requiring knowledge of targe model. Meng *et.al.* tried to achieve multi-target attacks with adding different types of signals to clean EEG, *i.e.*, NPP, sawtooth, sine, and chirp (Meng et al., 2023). However, these signals are not stealthy in both the temporal and frequency domain.

Different from the EEG BA in the temporal domain, we firstly propose to attack in the frequency domain. Our attack is **1)** more stealthy than NPP-based attack, **2)** faster than other trigger generation attack, and **3)** more practical as requiring no control of the target models. Compared to those frequency-based BAs for image, our attack introduces reinforcement learning to find the optimal injection strategy and design two novel rewards for enhancing the stealthiness and robustness.

It is worth noting that the adversarial attack (AA) (Zhang & Wu, 2019; Liu et al., 2021) is different from BA. AA tries to make the target model misclassify by adding invisible perturbation to input, which acts in the inference stage. BA tries to inject backdoor into target model in the training stage.

## 2.3 BACKDOOR DEFENSES

To cope with the security problems of backdoor attacks, several categories of defensive methods have been developed. Neural Cleanse (Wang et al., 2019) is a trigger reconstruction based methods. If the reconstructed trigger pattern is significantly small, the model is identified as a backdoor model. Assuming the trigger is still effective when a triggered sample is combining with a clean sample, STRIP (Gao et al., 2019) detects the backdoor model by feeding the combined samples into the model to see if the predictions are still with low entropy. Spectral Signature (Tran et al., 2018) detects the backdoor model based on the latent representations. Fine-Pruning (Liu et al., 2018a) erases the backdoor by pruning the model.

Besides the above defenses designed for backdoor attacks, there are some common EEG pre-processing methods, such as bandstop filtering and down-sampling, should be considered when designing a practical robust backdoor attack for EEG BCI in the real-world scene.

## 3 METHODOLOGY

### 3.1 EEG BCI BACKDOOR ATTACKS AND THREAT MODEL

**Multi-target BA.** The main notations in this paper are listed in Table 7. Under the supervised learning setting, a classifier $f$ is learned using a labeled training set $\mathcal{S} = \{(x_1, y_1), ..., (x_N, y_N)\}$ to map $f : \mathcal{X} \to \mathcal{C}$, where $x_i \in \mathcal{X}$ and $y_i \in \mathcal{C}$. The attacker in single target class backdoor attacks aims to learn a classifier $f$ behaves as follows:

$$f(x_i) = y_i, \ \ f(\mathcal{T}(x_i)) = c_{tar}, \ \ c_{tar} \in \mathcal{C}, \ \forall (x_i, y_i) \in \mathcal{S}, \tag{1}$$

where $\mathcal{T} : \mathcal{X} \to \mathcal{X}$ is the trigger function and $c_{tar}$ is the target label. For multi-target class backdoor attacks, the trigger function has an extra parameter $c_i$, which manipulates the behavior of $f$ flexibly:

$$f(x_i) = y_i, \;\; f(\mathcal{T}(c_i, x_i)) = c_i, \;\; \forall c_i \in \mathcal{C}, \forall (x_i, y_i) \in \mathcal{S}. \tag{2}$$

**Threat Model.** We consider a malicious data provider, who generates a small number of poisoned samples (labeled with the target class) and injects them into the original dataset. A victim developer collects this poisoned dataset and trains his model, which will be infected a backdoor.

## 3.2 REINFORCEMENT LEARNING FOR OPTIMAL TRIGGER-INJECTION STRATEGIES

The learning of the injecting electrodes set $\mathcal{M}_e^{c_i}$ and frequencies set $\mathcal{M}_f^{c_i}$ for each selected trigger in class $c_i$ can be formulated as a non-convex optimization problem. Under this optimization framework, the strategy generator function learn the optimal $\mathcal{M}_e^{c_i}$ and $\mathcal{M}_f^{c_i}$ for each EEG trigger to implement Professor X on target EEG BCI $f$, which is supposed to have a high clean accuracy (CA) on the clean data and attack success rate (ASR) on the poisoned data:

$$\underset{\mathcal{M}_e^{c_i}, \mathcal{M}_f^{c_i}}{\arg\min} \; \mathbb{E}_{(x_i, y_i) \sim \mathcal{D}}[\mathcal{L}(f(x_i), y_i) + \lambda \mathcal{L}(f(\mathcal{T}(x_i, x_{c_i}^t, \alpha, \mathcal{M}_e^{c_i}, \mathcal{M}_f^{c_i})), c_i)], \tag{3}$$

where $\lambda$ is a hyper-parameter to balance CA and ASR, and $\mathcal{T}$ is the poisoned data generation function. However, it is infeasible to find the optimal injecting strategy for each trigger in a large searching space, *e.g.*, if injecting half of the 62 electrodes, there are $\binom{62}{31} \approx 4.65 \times 10^{17}$ cases for deciding $\mathcal{M}_e^{c_i}$. Reinforcement learning (RL) is an appropriate method, whose objective of RL is to find a sampler $\pi$ to maximize the expect of the reward function. The details are presented in Algorithm 1.

$$\pi^* = \arg\max_\pi \mathbb{E}_{\tau \sim \pi(\tau)}[R(\tau)]$$
$$= \arg\max_\pi \sum_\tau [R(\tau) \cdot p_\pi(\tau)]$$
$$= \arg\max_\pi \sum_\tau [R(\tau) \cdot \rho_0(s_1) \cdot \tag{4}$$
$$\prod_{t=1}^{T-1} \pi(a_t | s_t) \cdot \mathcal{P}(s_{t+1} | s_t, a_t)],$$

where $R(\tau)$ is reward function of a trajectory $\tau = (s_1, a_1, r_1, .... s_T)$, the $s_i, a_i, r_i$ means the state, action, and reward at time $i$. The $\rho_0$ indicates the sampler of initial state. In our settings, the action (strategy) do not affect the state (trigger), which allows us to simplify Eq 4 by removing the states $s_i$:

$$\pi^* = \arg\max_\pi \sum_\tau [R(\tau) \cdot \prod_{t=1}^{T-1} \pi(a_t)]. \tag{5}$$

---

**Algorithm 1** Professor X's Strategy Optimization
**Input:** (1) dataset $\mathcal{S} = \{\mathcal{D}_{train}, \mathcal{D}_{test}, \mathcal{D}_p\}$,
    (2) trigger EEG $x_c^t$, policy network $\pi_\theta^c$,
    (3) iterations $K$ to update $\pi_\theta^c$,
    (4) poisoning function $\mathcal{T}$ (in section 3.3)
**Output:** learned strategies $\mathcal{M}_e^c$ and $\mathcal{M}_f^c$.
1: Initialize parameters $\theta$, $j \leftarrow 0$, $R_{best} \leftarrow 0$
2: **repeat**
3:     Sample two strategies: $\hat{\mathcal{M}}_e^c, \hat{\mathcal{M}}_f^c \leftarrow \pi_\theta(x_c^t)$
4:     Initialize poisoning set $\mathcal{S}_p \leftarrow \{\}$
5:     **for** each $(x_i, y_i) \in \mathcal{D}_p$ **do**
6:       **if** $y_i == c$ **then**
7:         $x_i^p \leftarrow \mathcal{T}(x, x_c^t, \alpha, \hat{\mathcal{M}}_e^c, \hat{\mathcal{M}}_f^c)$
8:         $\mathcal{S}_p \leftarrow \mathcal{S}_p + x_i^p$
9:       **end if**
10:     **end for**
11:     Train an EEG BCI on the set $\{\mathcal{D}_{train}, \mathcal{S}_p\}$
12:     Calculate CA and ASR on $\mathcal{D}_{test}$
13:     $R_t(\hat{\mathcal{M}}_e^{c_i}, \hat{\mathcal{M}}_f^{c_i}) \leftarrow$ CA $+ \lambda$ ASR $+ \mu \operatorname{dis}(\hat{\mathcal{M}}_f^{c_i}) + \nu \min(\hat{\mathcal{M}}_f^{c_i})$
14:     $\hat{g} \leftarrow \mathbb{E}_t[R_t(a_t) \cdot \nabla_\theta \log \pi_\theta]$
15:     Update $\theta$ with gradient $\hat{g}$: $\theta \leftarrow \theta + \eta\hat{g}$
16:     **if** $R_t(\hat{\mathcal{M}}_e^{c_i}, \hat{\mathcal{M}}_f^{c_i}) > R_{best}$ **then**
17:       $R_{best} \leftarrow R_t(\hat{\mathcal{M}}_e^{c_i}, \hat{\mathcal{M}}_f^{c_i})$
18:       $\mathcal{M}_e^c \leftarrow \hat{\mathcal{M}}_e^c, \mathcal{M}_f^c \leftarrow \hat{\mathcal{M}}_f^c$
19:     **end if**
20:     $j \leftarrow j + 1$
21: **until** $j = K$
22: **return** $\mathcal{M}_e^c, \mathcal{M}_f^c$

---

Furthermore, since only a particular strategy of each trigger matters, we replace the $R(\tau)$ with $R(a_t)$ and select the $a_t$ whose $R(a_t)$ is the biggest as the optimal strategy. Here, an RL algorithm called policy gradient (Sutton et al., 1999) is adopted to learn an agent (*i.e.*, policy network $\pi_\theta^{c_i}$ with parameters $\theta$) to find the optimal strategy for each trigger from class $c_i$. After removing the state $s_t$ and replacing $R(\tau)$, the gradient estimator is:

$$\hat{g} = \nabla_\theta \mathbb{E}_{\tau \sim \pi_\theta(\tau)}[R(\tau)] = \sum_\tau [R(a_t) \cdot \nabla p_{\pi_\theta}(a_t)] = \mathbb{E}_t[R_t(a_t) \cdot \nabla_\theta \log \pi_\theta], \tag{6}$$

where $a_t$ and $R_t$ is the action and estimator of the reward function at timestep $t$. The expectation $\mathbb{E}_t$ indicates the empirical average. Here, $a_t = \{\mathcal{M}_e^{c_i}, \mathcal{M}_f^{c_i}\}$. The parameters of $\pi_\theta^{c_i}$ are updated by $\theta_{t+1} = \theta_t + \eta\hat{g}$, $\eta$ is the learning rate. We run the RL for $K$ steps and take the best $a_t$ as the strategy.

Specifically, the agent has two output vectors $v_1 \in \mathbb{R}^E, v_2 \in \mathbb{R}^F$, where $E$ and $F$ is the number of EEG electrodes and frequencies. The electrodes and frequencies are in $\mathcal{M}_e^{c_i}$ and $\mathcal{M}_f^{c_i}$ only if the corresponding positions in $v_1$ and $v_2$ have Top-$k$ values, $k$ is $\gamma E$ for electrodes and $\beta F$ for frequencies, where $\gamma, \beta \in (0, 1]$ are hyperparameters.

Besides the CA and ASR, two other important concerns should be considered: **C1:** Robustness against common EEG preprocessig-based defenses. For instance, if a BA's trigger is injected into frequency band 50-60Hz, the BA will fail when EEG is filtered by a 50Hz low pass filter. Thus, scattering the injection positions in various frequency can effectively evade from specific frequency filter preprocessing. **C2:** Stealthiness against human perceptions. Since high frequency are related to environmental noise, injecting higher frequencies is more invisible (Gliske et al., 2016). Therefore, we design two novel loss functions to address **C1** and **C2**, DIS for scattering injection positions and HF for injecting higher frequencies. The whole reward function $R_t$ can be formulated follows:

$$R_t(a_t) = R_t(\mathcal{M}_e^{c_i}, \mathcal{M}_f^{c_i}) = \mathrm{CA} + \lambda \, \mathrm{ASR} + \mu \, \mathrm{dis}(\mathcal{M}_f^{c_i}) + \nu \min(\mathcal{M}_f^{c_i}), \qquad (7)$$

where the $\mathcal{M}_f^{c_i}$ indicates the set of all injecting frequency positions, and $\mathrm{dis}()$ calculates the minimal distance between each pair of positions. Thus, $\mathrm{dis}(\mathcal{M}_f^{c_i})$ is the discrete (DIS) loss, and $\min(\mathcal{M}_f^{c_i})$ is the high frequency (HF) loss, which can scatter the injection positions in various frequency bands and inject as high frequencies as possible. The $\lambda, \mu, \nu \in \mathbb{R}$ are hyperparameters.

### 3.3 Poisoned Data Generation in the Frequency Domain

After selecting the $C$ triggers from each class and learning the strategy for each trigger, the poisoned data are generated by injecting these triggers into clean data with the corresponding strategies. As shown in Fig 1(c), given a clean data $x_i \in \mathcal{D}_p$ with label $c_i$, and a trigger data $x_{c_i}^t$, let $\mathcal{F}^A$ and $\mathcal{F}^P$ be the amplitude and phase components of the fast Fourier transform (FFT) result of a EEG signals, we denote the amplitude and phase spectrum of $x_i$ and $x_{c_i}^t$ as:

$$\mathcal{A}_{x_i} = \mathcal{F}^A(x_i), \mathcal{A}_{x_{c_i}^t} = \mathcal{F}^A(x_{c_i}^t),$$
$$\mathcal{P}_{x_i} = \mathcal{F}^P(x_i), \mathcal{P}_{x_{c_i}^t} = \mathcal{F}^P(x_{c_i}^t). \qquad (8)$$

The new poisoned amplitude spectrum $\mathcal{A}_{x_i}^P$ is produced by linearly interpolating $\mathcal{A}_{x_i}$ and $\mathcal{A}_{x_{c_i}^t}$. In order to achieve this, we produce a binary mask $\mathcal{M}^{c_i} \in \mathbb{R}^{E \times F} = 1_{(j,k)}, j \in \mathcal{M}_e^{c_i}, k \in$

---

**Algorithm 2** Frequency Injection of Professor X: $\mathcal{T}(x, x_c^t, \alpha, \mathcal{M}_e^c, \mathcal{M}_f^c)$

**Input:** (1) clean EEG $x$, trigger EEG $x_c^t$ from class $c$, interpolating ratio $\alpha$,
(2) learned strategies $\mathcal{M}_e^c, \mathcal{M}_f^c$.
**Output:** the poisoned EEG $x^p$.
1: $\mathcal{M}^c \leftarrow$ a zero matrix with the shape of $E \times F$
2: **for** each $i \in \mathcal{M}_e^c$ **do**
3:     **for** each $j \in \mathcal{M}_f^c$ **do**
4:         $\mathcal{M}^c[i, j] \leftarrow 1$
5:     **end for**
6: **end for**
7: $\mathcal{A}_x, \mathcal{P}_x, \mathcal{A}_{x_c^t} \leftarrow \mathcal{F}^A(x), \mathcal{F}^P(x), \mathcal{F}^A(x_c^t)$
8: $\mathcal{A}_x^P \leftarrow [(1 - \alpha)\mathcal{A}_x + \alpha \mathcal{A}_{x_c^t}] \odot \mathcal{M}^c + \mathcal{A}_x \odot (1 - \mathcal{M}^c)$
9: $x^p \leftarrow \mathcal{F}^{-1}(\mathcal{A}_x^P, \mathcal{P}_x)$
10: **return** $x^p$

---

$\mathcal{M}_f^{c_i}$, whose value is 1 for all positions corresponding to elements in both electrode and frequency strategies and 0 elsewhere. Denoting $\alpha \in (0, 1]$ as the linear interpolating ratio, the new poisoned amplitude spectrum can be computed as follows, where $\odot$ indicates Hadamard product:

$$\mathcal{A}_{x_i}^P = [(1 - \alpha)\mathcal{A}_{x_i} + \alpha \mathcal{A}_{x_{c_i}^t}] \odot \mathcal{M}^{c_i} + \mathcal{A}_{x_i} \odot (1 - \mathcal{M}^{c_i}). \qquad (9)$$

Finally, we adopt the injected poisoned amplitude spectrum $\mathcal{A}_{x_i}^P$ and the clean phase spectrum $\mathcal{P}_{x_i}$ to get the poisoned data by inverse FFT $\mathcal{F}^{-1}$: $x_i^p = \mathcal{F}^{-1}(\mathcal{A}_{x_i}^P, \mathcal{P}_{x_i})$. The detailed procedure is written in Algorithm 2. By generating $x_i^p$ through this frequency injection approach, we obtain a subset $\mathcal{S}_p = \{x_1^p, ..., x_M^p\}$, which will combine with $\mathcal{D}_{train}$ to form the whole traing dataset $\mathcal{S}$. The EEG BCI model $f$ is then trained with $\mathcal{S}$ to obtain the ability of behvaing as equation 2.

## 4 Experiment Settings

### 4.1 Datasets

We demonstrate the effectiveness and generalizability of the proposed Professor X backdoor through comprehensive experiments on three EEG datasets. Some meta information is displayed in Table 1,

where can be seen that these datasets vary significantly in tasks, electrode numbers, montages, and sampling rates. More details about preprocessing are illustrated in Appendix E. Our goal is to develop a task-agnostic and format-agnostic BA method for EEG BCI. Hence, these elaborately chosen datasets can effectively validate the generalizability of each BA method.

Table 1: Meta information of the three datasets

| Dataset | # Class | # Subject | # Electrode | Sampling Rate | Montage |
|---|---|---|---|---|---|
| Emotion Recognition | 3 | 15 | 62 | 200 Hz | unipolar |
| Motor Imagery | 4 | 9 | 22 | 250 Hz | unipolar |
| Epilepsy Detection | 4 | 23 | 23 | 256 Hz | bipolar |

**Emotion Recognition (ER) Dataset.** SEED (Zheng & Lu, 2015) is a discrete EEG emotion dataset studying three types of emotions: happy, neutral, and sad. SEED collected EEG from 15 subjects.

**Motor Imagery (MI) Dataset.** BCIC-IV-2a (Brunner et al., 2008) dataset recorded EEG from 9 subjects while they were instructed to imagine four types of movements: left hand, right hand, feet, and tongue.

**Epilepsy Detection (ED) Dataset.** CHB-MIT (Shoeb & Guttag, 2010) is an epilepsy dataset required from 23 patients. We cropped and resampled the CHB-MIT dataset to build an ED dataset with four types of EEG: ictal, preictal, postictal, and interictal phase EEG.

## 4.2 BASELINES

**Non-stealthy Baselines.** As mentioned in previous sections, to the best of our knowledge, Professor X is the first work that studies multi-trigger and multi-target class (MT) backdoor in EEG BCI. For comparison, we design several baseline approaches which can be divided into two main groups: non-stealthy and stealthy. Non-stealthy attacks contains **PatchMT** and **PulseMT**. For a benign EEG segment $x \in \mathbb{R}^{E \times T}$. PatchMT is a multi-trigger and MT extension of BadNets (Gu et al., 2019) where we fill the first $\beta T$ timepoints of a EEG segments with a constant number, *e.g.*, $\{0.1, 0.3, 0.5\}$ for three-class task. PulseMT is a multi-trigger and MT extension of NPP-based backdoor attacks (Meng et al., 2023) where we use NPP signals with different amplitudes, *e.g.*, $\{-0.8, -0.3, 0.3, 0.8\}$ for different target classes.

**Stealthy Baselines.** Previous works generate stealthy poisioned samples by controlling the training stage and can only attack single target class (Meng et al., 2024; Ding et al., 2022; Jiang et al., 2023c). As they control the training of target model, it is unfair to directly compare their methods with Professor X. There is no stealthy MT BA for EEG. Thus, we design two MT stealthy attacks baselines: **CompMT** and **AdverMT**. CompMT generates poisoned samples for different target classes by compressing the amplitude of EEG with different ratios, *e.g.*, $\{-0.1, 0, 0.1\}$ for three-class task. AdverseMT is a multi-trigger and MT extension of adversarial filtering based attacks (Meng et al., 2024), where we using a local model trained only on $\mathcal{S}_p$ to generate different spatial filters $\mathbf{W}_i^*$ for different target classes, then we apply these spatial filters to generate poisoned samples. More details are written in Appendix F.

## 4.3 EXPERIMENTAL SETUP

We follow the poisoning attack setting as the previous works (Meng et al., 2023) and consider three widely-used EEG BCIs for classifier $f$: EEGNet (Lawhern et al., 2018), DeepCNN (Schirrmeister et al., 2017), and LSTM (Tsiouris et al., 2018). We use a cross-validation setting to evaluate all BAs, each EEG dataset $\mathcal{D}$ is divided into three parts: training set $\mathcal{D}_{train}$, poisoning set $\mathcal{D}_p$, and test set $\mathcal{D}_{test}$. Specifically, for a dataset contains $n$ subjects, we select one subject's data as $\mathcal{D}_p$ one by one, and the remaining $n-1$ subjects to perform leave-one-subject-out (LOSO) cross-validation, *i.e.*, one of the subjects as $\mathcal{D}_{test}$, and the remaining $n-2$ subjects as $\mathcal{D}_{train}$ (one of the subjects in $\mathcal{D}_{train}$ is chosen to be validation set). In summary, for a dataset contains $n$ subjects, there are $n(n-1)$ runs to validate each EEG BCI backdoor attack method. A poisoned subset $\mathcal{S}_p$ of $M$ ($M < N$) examples is generated based on $\mathcal{D}_p$. Then $\mathcal{S}_p$ is combined with $\mathcal{D}_{train}$ to acquire $\mathcal{S} = \{\mathcal{S}_p, \mathcal{D}_{train}\}$. The poisoning ratio is defined as : $\rho = M/N$.

Table 2: The clean accuraciy and attack success rate for each target class with 40% poisoning rate. The best results are in **bold** and the second best are underlined. (M1: TimesNet, M2: EEG-Conformer)

| | Dataset | Emotion Recognition | | | | | Motor Imagery | | | | | | Epilepsy Detection | | | | | |
|---|---|---|---|---|---|---|---|---|---|---|---|---|---|---|---|---|---|---|
| | Method | Clean | ASR | 0 | 1 | 2 | Clean | ASR | 0 | 1 | 2 | 3 | Clean | ASR | 0 | 1 | 2 | 3 |
| **EEGNet** | No Attack | 0.477 | 0.333 | - | - | - | 0.327 | 0.250 | - | - | - | - | 0.508 | 0.250 | - | - | - | - |
| | PatchMT | 0.492 | 0.382 | 0.577 | 0.232 | 0.337 | 0.283 | 0.824 | 0.866 | 0.880 | 0.787 | 0.762 | 0.460 | 0.549 | 0.532 | 0.430 | 0.388 | 0.845 |
| | PulseMT | 0.463 | 0.778 | **0.844** | 0.509 | **0.981** | 0.270 | 0.825 | 0.947 | 0.656 | 0.758 | 0.938 | 0.439 | 0.810 | 0.853 | 0.745 | 0.729 | 0.913 |
| | CompMT | 0.443 | 0.385 | 0.099 | 0.377 | 0.678 | 0.269 | 0.865 | 0.530 | 0.997 | 0.983 | 0.948 | 0.437 | 0.547 | 0.261 | 0.280 | 0.714 | 0.933 |
| | AdverMT | 0.457 | 0.334 | 0.276 | 0.330 | 0.396 | 0.257 | 0.243 | 0.316 | 0.192 | 0.230 | 0.235 | 0.413 | 0.250 | 0.326 | 0.264 | 0.200 | 0.210 |
| | Professor X | **0.535** | **0.857** | 0.831 | **0.791** | 0.949 | **0.323** | **1.000** | **0.999** | **1.000** | **1.000** | **0.999** | **0.477** | **0.944** | **0.930** | **0.954** | **0.921** | **0.970** |
| **DeepCNN** | No Attack | 0.497 | 0.333 | - | - | - | 0.301 | 0.250 | - | - | - | - | 0.443 | 0.250 | - | - | - | - |
| | PatchMT | 0.481 | 0.342 | 0.248 | 0.323 | 0.453 | 0.276 | 0.704 | 0.638 | 0.977 | 0.774 | 0.425 | 0.431 | 0.729 | 0.416 | **0.890** | 0.719 | 0.892 |
| | PulseMT | 0.450 | 0.596 | **0.815** | 0.334 | 0.638 | 0.261 | 0.829 | 0.764 | 0.968 | 0.819 | 0.765 | 0.405 | **0.885** | **0.872** | 0.862 | **0.861** | **0.943** |
| | CompMT | 0.461 | 0.427 | 0.473 | 0.473 | 0.336 | 0.286 | 0.887 | 0.638 | 0.982 | 0.946 | 0.980 | 0.446 | 0.538 | 0.196 | 0.466 | 0.571 | 0.918 |
| | AdverMT | 0.367 | 0.388 | 0.298 | 0.453 | 0.412 | 0.245 | 0.247 | 0.320 | 0.221 | 0.196 | 0.240 | 0.396 | 0.275 | 0.354 | 0.218 | 0.227 | 0.301 |
| | Professor X | **0.534** | **0.832** | 0.732 | **0.865** | **0.901** | **0.315** | **1.000** | **1.000** | **1.000** | **1.000** | **0.999** | **0.469** | 0.828 | 0.725 | 0.839 | 0.845 | 0.904 |
| **LSTM** | No Attack | 0.506 | 0.333 | - | - | - | 0.264 | 0.250 | - | - | - | - | 0.462 | 0.250 | - | - | - | - |
| | PatchMT | 0.509 | 0.368 | 0.311 | 0.392 | 0.401 | 0.261 | 0.429 | 0.395 | 0.296 | 0.386 | 0.639 | 0.450 | 0.513 | 0.500 | 0.437 | 0.417 | 0.700 |
| | PulseMT | 0.511 | 0.824 | 0.883 | 0.645 | 0.943 | **0.265** | 0.533 | 0.787 | 0.327 | 0.282 | 0.737 | 0.451 | 0.804 | **0.845** | 0.769 | 0.709 | 0.895 |
| | CompMT | 0.484 | 0.490 | 0.272 | 0.269 | 0.929 | 0.260 | 0.548 | 0.219 | 0.511 | 0.523 | 0.940 | **0.455** | 0.435 | 0.194 | 0.217 | 0.490 | 0.840 |
| | AdverMT | 0.367 | 0.415 | 0.472 | 0.453 | 0.321 | 0.239 | 0.271 | 0.308 | 0.215 | 0.247 | 0.312 | 0.432 | 0.268 | 0.367 | 0.232 | 0.198 | 0.275 |
| | Professor X | **0.519** | **0.954** | **0.998** | **0.868** | **0.996** | 0.264 | **0.966** | **0.987** | **0.988** | **0.901** | **0.986** | 0.444 | **0.865** | 0.795 | **0.833** | **0.857** | **0.975** |
| M1 | Professor X | 0.485 | 0.960 | 0.961 | 0.926 | 0.993 | 0.276 | 0.997 | 0.999 | 0.998 | 0.999 | 0.992 | 0.373 | 0.986 | 0.985 | 0.986 | 0.995 | 0.976 |
| M2 | Professor X | 0.475 | 0.894 | 0.842 | 0.904 | 0.935 | 0.935 | 0.996 | 0.999 | 1.000 | 0.987 | 0.999 | 0.419 | 0.944 | 0.958 | 0.970 | 0.887 | 0.964 |

For all methods, we train the classifiers using the Adam optimizer with learning rate of 0.001. The batch size is 32 and the number of epochs is 100. For all datasets and baselines, the interpolating ratio $\alpha = 0.8$, the frequency poisoning ratio $\beta = 0.1$, the electrode poisoning ratio $\gamma = 0.5$. For the reinforcement learning, we train $\pi_\theta$ networks $K = 250$ epochs using the Adam optimizer with learning rate of 0.01. The hyperparameters in advantage function is set to $\lambda = 2$, $\mu = 0.3$, and $\nu = 0.005$. More details of the experimental setup can be found in Appendix F.

## 5 EXPERIMENTAL RESULTS

### 5.1 EFFECTIVENESS OF PROFESSOR X

This section presents the attack success rates of Professor X and baselines. To evaluate the performance in the multi-trigger multi-payload scenario, for each test sample $(x, y) \in \mathcal{D}_{test}$, we enumerate all possible target labels $c_i \in \mathcal{C}$ including the true label $y$ and inject the trigger to activate the backdoor. The attack is successful only when the backdoor classifier $f$ correctly predicts $c_i$ for each poisoned input $x$ with a target label $c_i$.

#### 5.1.1 ATTACK PERFORMANCE

The CA (Clean) and ASR (Attack) for each class of all attack methods on three EEG tasks with three EEG BCI models are presented in Table 2. The AdverMT, designed for single-target attack, fails to attacks multiple target classes. While PulseMT achieves the second best on ER and ED dataset, CompMT achieves the second best on the MI dataset, indicating that these baselines are less generalizable. Our Professor X significantly outperforms baselines at almost all cases ($p < 0.05$) except attacking DeepCNN on the ED dataset, having ASRs above 0.8 on three datasets and even achieving an ASR of 1.000 on the MI dataset. Moreover, our attack is also effective on the SOTA time-reries classification model TimesNet (M1) (Wu et al., 2023) and Transformer-based model EEG-Conformer (M2) (Song et al., 2022). These results demonstrate that our Professor X is effective across different EEG tasks and EEG models, showcasing it's generalizability.

#### 5.1.2 PERFORMANCE OF THE REINFORCEMENT LEARNING: POLICY GRADIENT

Displaying in Table 3, the performance of the policy gradient was compared with other common optimazation algorithms, including genetic algorithm (GA) (Katoch et al., 2021) and random selection (The search space is too large for performing grid search as explained in Section 3.2). It can be observed that the policy gradient outperforms GA while only spending 16% training time of GA. We plot the learning curve of RL in Appendix H.3, which demonstrates that RL learns well strategies within 50 epochs, i.e., only trains 50 backdoor models and saves lots of time. It is worth mentioning

that the random algorithm achieves not bad results, proving that our methods can be applied without RL if some performance drop is acceptable.

Table 3: Clean and attack performance with with different trigger search optimization algorithms, the poisoning rate is set to 10%. The target model is EEGNet.

| Dataset / Method | Emotion | | | Motor Imagery | | | Epilepsy | | |
|---|---|---|---|---|---|---|---|---|---|
| | Clean | Attack | Time ↓ | Clean | Attack | Time ↓ | Clean | Attack | Time ↓ |
| Random | 0.520 | 0.771 | - | 0.291 | 0.857 | - | 0.501 | 0.721 | - |
| Genetic Algorithm | 0.516 | 0.826 | 15.2h | 0.302 | 1.000 | 10.0h | 0.492 | 0.862 | 30.5h |
| Policy Gradient | 0.535 | 0.857 | 2.5h | 0.323 | 1.000 | 1.8h | 0.477 | 0.944 | 5.2h |

### 5.1.3 PERFORMANCE OF LEARNED MASK STRATEGIES ON OTHER TARGET MODELS

We demonstrate that the injecting strategies learned on a EEG classifier $f$ can be used to attack other EEG classifiers $\hat{f}$. In other words, Professor X can still be effective when the adversary has no knowledge of the target models $\hat{f}$. To perform the experiments, we use the strategy learned with a classifier $f$, then generate poisoned samples to attack another classifier $\hat{f}$ whose network is different from $f$. Table 4 shows the performance difference, it can be observed that the difference is relatively small in most of the cases, demonstrating the transferability of the injecting strategy learned with reinforcement learning.

Table 4: Clean and attack performance on other models. Red values represent the decreasing performance in attacks with $f$ is the same as $\hat{f}$. Blue values mean increments or unchanged .

| Models | $f$ : EEGNet | | | | $f$ : DeepCNN | | | | $f$ : LSTM | | | |
|---|---|---|---|---|---|---|---|---|---|---|---|---|
| | $\hat{f}$ : DeepCNN | | $\hat{f}$ : LSTM | | $\hat{f}$ : EEGNet | | $\hat{f}$ : LSTM | | $\hat{f}$ : EEGNet | | $\hat{f}$ : DeepCNN | |
| Datasets | Clean | Attack | Clean | Attack | Clean | Attack | Clean | Attack | Clean | Attack | Clean | Attack |
| Emotion | 0.458 | 0.781 | 0.485 | 0.938 | 0.516 | 0.813 | 0.490 | 0.936 | 0.516 | 0.863 | 0.497 | 0.779 |
| | 0.026 | 0.051 | 0.034 | 0.016 | 0.019 | 0.044 | 0.029 | 0.018 | 0.019 | 0.006 | 0.037 | 0.053 |
| Motor | 0.316 | 1.000 | 0.265 | 0.946 | 0.309 | 1.000 | 0.264 | 0.972 | 0.306 | 1.000 | 0.306 | 1.000 |
| | 0.001 | 0.000 | 0.001 | 0.020 | 0.014 | 0.000 | 0.000 | 0.006 | 0.017 | 0.000 | 0.009 | 0.000 |
| Epilepsy | 0.442 | 0.759 | 0.469 | 0.806 | 0.448 | 0.943 | 0.445 | 0.813 | 0.448 | 0.926 | 0.427 | 0.850 |
| | 0.027 | 0.069 | 0.025 | 0.059 | 0.029 | 0.001 | 0.001 | 0.052 | 0.029 | 0.018 | 0.042 | 0.022 |

### 5.1.4 ATTACK PERFORMANCE WITH DIFFERENT HYPERPARAMETERS

We investigate the influences of three different hyperparameters: poisoning rate $\rho$, frequency injection rate $\beta$, and electrode injection rate $\gamma$. The performance of attacking EEGNet on the ED dataset are displayed in Fig 3. It can be seen that the ASRs are positively correlated with poisoning rate. Note that it is non-trivial for multi-target class attack, thus the ASR is not high compared to the single class attack. Professor X outperforms other attacks in all cases and is robust to the change of $\beta$ and $\gamma$.

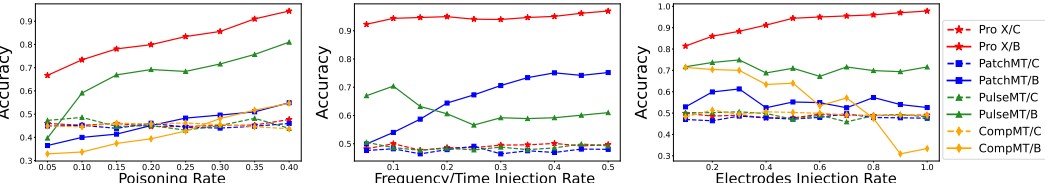

Figure 3: Clean (/C) and attack (/B) performance with different poisoning or injection rates.

## 5.2 ROBUSTNESS OF PROFESSOR X

In this section, we evaluate the robustness of our Professor X against different EEG preprocessing method and various representative backdoor defenses.

### 5.2.1 ROBUSTNESS AGAINST EEG PREPROCESSING METHODS

To develop an EEG BCI, it is very common to preprocess the raw EEG signals, *e.g.*, 1) band-stop filtering and 2) down-sampling. An EEG backdoor attack is impractical in real scenarios if it is no longer effective when the target model is trained with the preprocessed poisoned EEG. Hence, we must take the robustness against preprocessing methods into account, which is widely ignored in the image backdoor attack field. The performance of each method facing different preprocessing methods are presented in Table 5. It can be observed that our Professor X is robust in all cases. However, when removing the DIS loss, the performance of Professor X decreases a lot after EEG preprocessing, especially facing the 30 Hz high-stop filtering preprocessing due to the HF loss that encourages the policy network learns to injecting high frequency.

Table 5: Clean and attack performance on three datasets after different EEG preprocessing methods. The target model is EEGNet. M w.o. DIS means removing the DIS loss in Professor X.

|  | Preprocessing | No defense | | 20 Hz low | | 30 Hz high | | 25% down | | Average |
|---|---|---|---|---|---|---|---|---|---|---|
|  | Method | Clean | Attack | Clean | Attack | Clean | Attack | Clean | Attack | ASR |
| ER | Professor X | 0.535 | 0.857 | 0.512 | 0.829 | 0.463 | 0.892 | 0.518 | 0.908 | 0.876 |
| ER | w/o DIS | 0.506 | 0.859 | 0.492 | 0.816 | 0.466 | 0.333 | 0.498 | 0.807 | 0.652 |
| MI | Professor X | 0.323 | 1.000 | 0.285 | 1.000 | 0.329 | 1.000 | 0.321 | 1.000 | 1.000 |
| MI | w/o DIS | 0.298 | 1.000 | 0.264 | 1.000 | 0.322 | 0.250 | 0.284 | 0.990 | 0.746 |
| ED | Professor X | 0.497 | 0.944 | 0.492 | 0.914 | 0.494 | 0.856 | 0.516 | 0.818 | 0.920 |
| ED | w/o DIS | 0.515 | 0.250 | 0.477 | 0.864 | 0.508 | 0.250 | 0.510 | 0.249 | 0.454 |

### 5.2.2 ROBUSTNESS AGAINST NEURAL CLEANSE: TRIGGER INVERSION

Neural Cleanse (NC) (Wang et al., 2019) calculate a metric called Anomaly Index by reconstructing trigger pattern for each possible label. The Anomaly Index is positively correlated with the size of the reconstruction trigger. A model with Anomaly Index > 2 is considered to be backdoor-injected. We display the Anomaly Indexes of the clean models and the backdoor-injected model by Professor X in Fig 4. It can be seen that Professor X can easily bypass NC. The reconstructed trigger patterns on three datasets are presented in Appendix H.1.

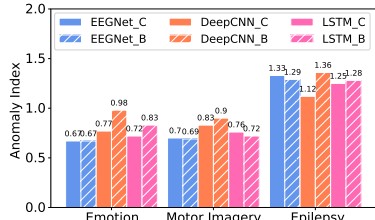

Figure 4: Anomaly Index of three models on three datasets.

### 5.2.3 ROBUSTNESS AGAINST STRIP: INPUT PERTURBATION

We evaluate the robustness of Professor X against STRIP (Gao et al., 2019), which perturbs the input EEG and calculates the entropy of the predictions of these perturbed EEG data. Based on the assumption that the trigger is still effective after perturbation, the entropy of backdoor input tends to be lower than that of the clean one. The results are plotted in Fig 5, it can be seen that the entropy distributions of the backdoor and clean samples are similar.

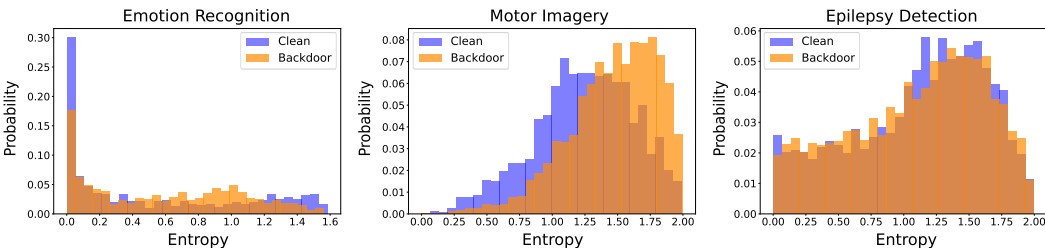

Figure 5: Performance against STRIP on three datasets, the target model is EEGNet.

### 5.2.4 ROBUSTNESS AGAINST SPECTRAL SIGNATURE: LATENT SPACE CORRELATION

Spectral Signature (Tran et al., 2018) detects the backdoor samples by statistical analysis of clean data and backdoor data in the latent space. Following the same experimental settings in (Tran et al.,

2018), we randomly select 5,000 clean samples and 500 Professor X backdoor samples and plot the histograms of the correlation scores in Fig 6. There is no clear separation between these two sets of samples, showing the stealthiness of Professor X backdoor samples in the latent space.

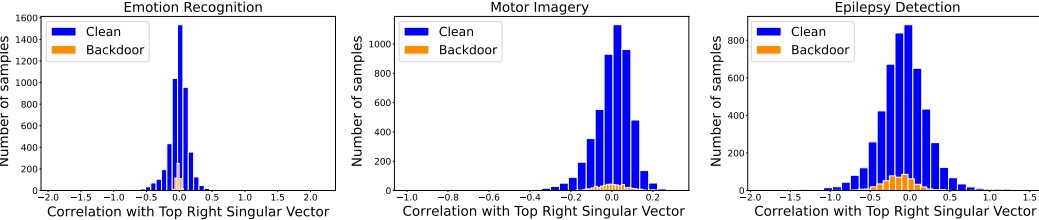

Figure 6: Performance against Spectral Signature on three datasets, the target model is EEGNet.

### 5.2.5 ROBUSTNESS AGAINST FINE-PRUNING

We evaluate the robustness of Professor X against Fine-Pruning (Liu et al., 2018a), a model analysis based defense which finds a classifier's low-activated neurons given a small clean dataset. Then it gradually prunes these low-activated neurons to mitigate the backdoor without affecting the CA. We can observe from Fig 7 that the ASR drops considerably small when pruning ratio is less than 0.7, suggesting that the Fine-Pruning is ineffective against Professor X.

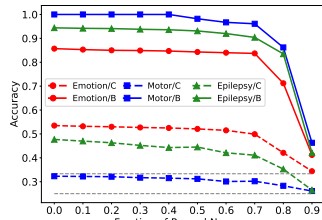

Figure 7: Performances of EEGNet against Fine-Pruning on three datasets.

### 5.3 VISUALIZATION OF BACKDOOR ATTACK SAMPLES

To evade from human perception (**C2** in Section 3.2), we design to obatin injecting strategies with HF loss. It can be seen from the bottom row of Fig 8 that Professor X (with HF loss) generates stealthy poisoned EEG, which is almost the same as the clean EEG, demonstrating the **High Stealthiness**. More visualization on three datasets are presented in Appendix H.2.

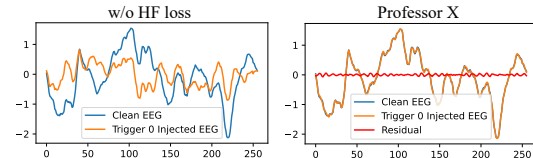

Figure 8: The Clean EEG (Blue), Trigger-injected EEG (Orange) and the Residual (Red) of the ED dataset. (*x*-axis: timepoints, *y*-axis: amplitude.)

### 5.4 STEALTHINESS AGAINST DETECTION

To verify that the trigger of Professor X are invisible, we employ anomaly detection methods, GDN (Deng & Hooi, 2021) and USAD (Audibert et al., 2020). Specifically, for each dataset, we train anomaly detection methods on the clean test set $\mathcal{D}_{test}$ and then record the F1-score and the Area under the ROC Curve (ROC-AUC) on the set $= \mathcal{S}_p \cup \mathcal{D}_p$. The experimental results are presented in Table 6. The ROC-AUC is around 0.5 and F1-score is either around 0.5 or near 0 across all datasets, indicating that the detection results are nearly random guess. These strongly demonstrates the stealthiness of Professor X.

Table 6: Results of anomaly detection.

| Anomaly Detection | ER | | MI | | ED | |
|---|---|---|---|---|---|---|
| | F1 | AUC | F1 | AUC | F1 | AUC |
| GDN | 0.50 | 0.50 | 0.50 | 0.51 | 0.50 | 0.50 |
| USAD | 0.00 | 0.51 | 0.00 | 0.51 | 0.00 | 0.50 |

## 6 CONCLUSION

In this paper, we proposed Professor X, a novel EEG backdoor for manipulating EEG BCI, where the adversary can arbitrarily control the output for any input samples. To the best of our knowledge, Professor X is the first method that considers which EEG electrodes and frequencies to be injected for different EEG tasks and formats. We specially design the reward function in RL to enhance the robustness and stealthiness. Experimental results showcase the effectiveness, robustness, and generalizability of Professor X. This work alerts the EEG community of the potential danger of the vulnerability of EEG BCI against BA and calls for defensive studies for EEG modality. It is worth noting that Professor X can also be applied for protecting intellectual properties of EEG datasets and BCI models, offering a concealed and harmless approach to add authors' watermark (backdoor can be regarded as watermark), indicating the real-world application of Professor X.

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

# A KEY SYMBOLS OF PROFESSOR X

In this section, we list all the key symbols used in our paper in Table 7.

Table 7: Key symbols.

| Symbol | Definition |
|---|---|
| $x_i$ | The input data |
| $y_i$ | The input data's label |
| $x_c^t$ | The randomly selected trigger from class $c$ (with label $c$) |
| $x_i^p$ | The poisoned data of input data $x_i$ |
| $c_{tar}$ | The target class in the single target class backdoor attacks |
| $E$ | The number of electrodes of an EEG segment |
| $F$ | The number of frequency points of an EEG segment after FFT |
| $T$ | The number of time points of an EEG segment |
| $N$ | The number of the data points in the training subset |
| $M$ | The number of the data points in the poisoning subset |
| $\alpha$ | The interpolating ratio of trigger and clean data |
| $\beta$ | The ratio of injection time/frequency points to total time/frequency points |
| $\gamma$ | The ratio of injection electrodes to total electrodes |
| $\lambda$ | The hyperparameter to balance the ASR reward in reinforcement learning |
| $\mu$ | The hyperparameter to balance the DIS loss in reinforcement learning |
| $\nu$ | The hyperparameter to balance the HF loss in reinforcement learning |
| $\rho$ | The ratio of the size of the poisoning subset to that of the training set |
| $\pi_\theta^{c_i}$ | The policy network for the selected trigger from class $c_i$ with parameter $\theta$) |
| $\theta$ | The parameter of the policy network |
| $\rho_0$ | The sampler of initial state |
| $s_i$ | The state at time point $i$ in reinforcement learning |
| $a_i$ | The action at time point $i$ in reinforcement learning |
| $r_i$ | The reward at time point $i$ in reinforcement learning |
| $\tau$ | The trajectory of the whole decision made by policy network |
| $\hat{g}$ | The gradient estimator of a reward taken by a trajectory |
| $\eta$ | The learning rate for training policy network |
| $R$ | The reward function of a trajectory or a single action |
| $K$ | The iteration numbers of reinforcement learning |
| $\mathcal{X}$ | The distribution of input data |
| $\mathcal{C}$ | The distribution of label/class |
| $\mathcal{M}_e^{c_i}$ | The injecting electrodes set of the selected trigger from class $c_i$ |
| $\mathcal{M}_f^{c_i}$ | The injecting frequencies set of the selected trigger from class $c_i$ |
| $\mathcal{M}^{c_i}$ | The binary mask of the selected trigger from class $c_i$ |
| $\mathcal{S}$ | The set of labeled training data |
| $\mathcal{T}$ | The trigger-injection function |
| $\mathcal{L}$ | The loss function used for training a classifier |
| $\mathcal{D}$ | The dataset used for training a classifier |
| $\mathcal{D}_{train}$ | The subset of $\mathcal{S}$ used for training the target model |
| $\mathcal{D}_{test}$ | The subset of $\mathcal{S}$ used for testing the target model |
| $\mathcal{D}_p$ | The subset of $\mathcal{S}$ used for poisoning the target model |
| $\mathcal{S}_p$ | The set of generated poisoned data using the subset $\mathcal{D}_p$ |
| $\mathcal{F}$ | The fast Fourier transform |
| $\mathcal{A}_x$ | The amplitude of data $x$ after FFT |
| $\mathcal{P}_x$ | The phase of data $x$ after FFT |
| $f$ | The target classifier model |
| $\hat{f}$ | The target model in the trigger transfer experiments |

## B LIMITATIONS

Our Professor X is a backdoor attack in the frequency domain, which requires to transform the EEG signals into frequency domain through fast Fourier transform (FFT) and return to temporal domain through inverse FFT (iFFT). The operation of FFT and iFFT in the trigger injection function are a little more time-consuming compared to other backdoor attack directly in the temporal domain, like PatchMT (Gu et al., 2019) and PulseMT (Meng et al., 2023). Future effort will be devoted into the faster implementation of FFT and iFFT, for example, taking the advantage of modern GPUs.

It is a little more time-consuming for the reinforcement learning to acquire the optimal strategies for each trigger. However, we can obtain a general injecting strategy for each EEG BCI tasks, which can achieve a relatively good performance without reinforcement learning, as we can see from Table 4 that random injection strategy has an acceptable performance.

## C BROADER IMPACTS

With the rapid development of techniques, EEG BCIs gain a wide range of applications from health care to human-computer interaction. Some companies like Neuralink adopt the EEG BCI to assist paralytic patients helping themselves in daily lives. However, if the EEG BCI is backdoor attacked by Professor X, which allows the attacker to arbitrarily control BCI's outputs, the BCI users may fall into tremendous fatal troubles. For instance, one paralytic patient controls his/her wheelchair by EEG BCI, the attacker can manipulate the wheelchair to run down a steep staircase. For an epileptic patient, the attacker can let all the output be Normal State, even when the patient is experiencing an epileptic seizure. This paper reveals the severe danger faced by EEG BCIs, demonstrating the possibility that someone can maliciously manipulate the outputs of EEG BCIs with arbitrary target class.

Professor X can also be used for positive purposes, like protecting intellectual property of EEG dataset and EEG models with watermarking. As our Professor X has a very small impact of the clean accuracy, and the poisoning approach is clean label poisoning, Professor X is a fantastic method for watermarking EEG dataset and models.

For a company that provides EEG dataset, it can select different EEG triggers for different customs to generate poisoned data and inject into the dataset provided to customs who buy the dataset. As a result, the company have the information of which trigger is corresponding to which customs, e.g., trigger $x$ is in the dataset provided to custom $X$, trigger $y$ is in the dataset provided to custom $Y$. If an EEG model from a company which didn't buy dataset is detected having this watermark (backdoor) with trigger $x$, the company knows that the custom $X$ leaked the dataset. Similarly, if an EEG model is detected having this watermark (backdoor) with trigger $y$, the company knows that the custom $Y$ leaked the dataset.

## D DISCUSSION OF DEFENSIVE STUDY AGAINST PROFESSOR X

Thanks to the reviewer JuBu, uEuL and kEdv in the ICLR conference, who asked many questions regarding the defensive study against Professor X. These insightful concerns deepen our understanding of our attack and how to guard backdoor attack in EEG BCIs. Thus, we add a new section here to discuss our humble opinion on the defensive study against Professor X, which we hope will benefit the future research.

Since backdoor attack is primarily studied in the image processing field, the defensive research is also conducted for protecting image model. However, EEG modality, a kind of multi-variate time series, is far different from image modality. These difference may inherently cause failure of existing backdoor defensive methods. Next, we would like to discuss the limitation of these defensive methods.

### D.1 NEURAL CLEANSE

Neural Cleanse (NC) (Wang et al., 2019) aims to reconstruct the trigger pattern in the backdoor model. It is conducted based on the following assumption:

1) The trigger pattern is the same for different input, which is called input-agnostic.

2) The backdoor model learns a *shortcut* for the trigger pattern.

3) The trigger pattern is relatively small compared to the whole input.

NC first initialize a random noise and a random noise as the trigger pattern, then optimize the noise and mask to make the backdoor model outputs the target label for a input injected with the trigger, and let the mask as small as possible. At last, NC calculate a anomaly index according to the size of the mask. The smaller the mask, the higher the anomaly index. Empirically, the anomly index threshold is set to 2. NC works well on detecting BA likes BadNets (Gu et al., 2019) and Trojan Backdoor (Liu et al., 2018b), which are basically consistent with the above assumptions.

However, the trigger patterns for EEG BCI are always not small, like NPP signals and our attack (the trigger can be seen in Fig. 8, the red residual is the trigger). These trigger patterns are wide and cover all time points of EEG signals. Thus, NC is not effective in detecting our attack. It can be seen from Fig. 9, 10 and 11 that the reconstructed trigger patterns of clean model and backdoor model are quite similar. And the mask reconstructed for both model are all very wide and extend to most channels and time points. In short, NC fails to detect our attack.

## D.2    STRIP

STRIP (Gao et al., 2019), which perturbs the input EEG and calculates the entropy of the predictions of these perturbed EEG data. STRIP detects the backdoor based on thees assumptions:

1) backdoor trigger is input-agnostic;

2) backdoor trigger is strong and effective when performing input perturbation;

3) the backdoor models' outputs (softmax) of poisoned data has very low entropy.

STRIP has several strengths:

1) **Insensitive to trigger-size**: STRIP is effective no matter the trigger is big or small.

2) **Plug and Play**: STRIP is plug and play, and compatible in any models. We only need the inputs and outpus of the backdoor models (treated as a black box as we don't need any intermediate outputs), then calculate the entropy of the outputs.

3) **Backdoor model architecture-agnositc**: STRIP only needs the inputs and outputs of the backdoor model, so it is an architecture-agnositc method and is generalize to many real-world application senarios.

However, STRIP also has some weaknesses. Any trigger that may affect the above findings may cause STRIP's detection failure:

1) the trigger is input-specific;

2) the trigger is not that strong, it fails when performing input perturbation;;

3) the trigger won't cause the backdoor model to predict with very low entropy.

So why STRIP fails in detecting Professor X? Firstly, our trigger is injected in the frequency domain, leading to the input-specific pattern in the temporal domain, causing assumption 1 to be invalid. Moreover, the input perturbation in the temporal domain may damages the frequency information, causing our trigger disapper, leading to the assumption 2 to be invalid. Lastly, as EEG is a nonstationary modality, the outputs of EEG models are always with high entropy, making assumption 3 to be invalid. Thus, STRIP is not effective in detecting Professor X attack.

## D.3    SPECTRAL SIGNATURE

Spectral Signature (Tran et al., 2018) detects the backdoor samples by statistical analysis of clean data and backdoor data in the latent space, which first find the top-right singular vector of the covariance matrix of the latent vectors of a small subset of clean samples, then each sample is calculated a correlation score to this singular vector. It detects whether a sample is backdoor sample by the

correlation score, the difference the correlation score, the higher the possibility of being a backdoor sample. Spectral Signature aims to purify the datasets, it can remove all the possible backdoor sample. However, any clean sample can also be possibly removed by Spectral Signature.

The reason of the failure of Spectral Signature on EEG BCI might be that EEG signals are nonstationary, so the latent space of EEG model contains a lots of noises. These noises causes the similarity between backdoor samples and clean samples.

### D.4 Fine-Pruning

Fine-Pruning (Liu et al., 2018a) assumes that the defender has a validation dataset $\mathcal{D}_{valid}$ in which all data are clean. The defender feeds these clean data into the backdoor models, and recrods the average activation of each neuron. Afterwards, the defender iteratively prunes neurons from the DNN in increasing order of average activations. Thus, the low-activated neurons are those the average activation is low when feeding in clean data.

However, Fine-Pruning can inadvertently remove important features that are crucial for classification. Because the average activation is obtained from the small subset $\mathcal{D}_{valid}$, so the low-activated neurons determined by $\mathcal{D}_{valid}$ may be high-activated neurons when feeding another clean validation dataset $\mathcal{D}'_{valid}$. That is, the important neurons for classifying clean sample $x \in \mathcal{D}'_{valid}$ may be low-activated neurons for all samples in $\mathcal{D}_{valid}$, resulting in the pruning of these important neurons.

As we discussed above, Fine-Pruning requires that the defender has a validation dataset As we discussed above, Fine-Pruning requires that the defender has a validation dataset. The performance of Fine-Pruning relies heavily on the quality of the validation dataset, since the low-activated neurons are determined by the validation dataset.

So in the future, building a large, diverse, high quality, and absolutely clean validation dataset is the key for improving the Fine-Pruning's performance. The most important part is the diversity, which not only means the diversity of EEG tasks, but also means the diversity of EEG formats. Thus, improving the defenses against backdoor attacks is not an easy task and needs joint efforts of the medical and academic communities.

### D.5 Anomaly Detection Method

Following the BackTime paper (Lin et al., 2024), we also conduct a same experiment. But for Professor X, the trigger is input-specific, resulting in these anomaly detection models does not see any trigger pattern before and thus cannot tell the whether a EEG data is a clean or backdoor sample.

## E Datasets and Preprocessing

In this section, we introduce the three datasets used in our experiments, and explain the preprocessing. We elaborately selected these three datasets because of three reasons: **1)** They cover three different EEG tasks that are important and common in EEG BCI field; **2)** The EEG formats of these datasets vary significantly; **3)** The EEG tasks are all multi class classification tasks, that is, the number of categories is more than two. Experiments on these three datasets can validate the efficacy, manipulating performance, and generalizability of each BA methods as much as possible.

### E.1 Emotion Recognition (ER)

The SJTU Emotion EEG Dataset (SEED) was incoporated as the representative dataset of emotion recogniton tasks (Zheng & Lu, 2015). It consists of EEG recordings from 15 subjects watching 15 emotional video clips with three repeated session each on different days. Each video clip is supposed to evoke one of the three target emotions: positive, neutral, and negative. The EEG signals were acquired by the 62-channel electrode cap at a sampling rate of 1000 Hz. We performed below preprocessing procedures for the 62-channel EEG signals: 1) Down-sampling from 1000 Hz to 200 Hz, 2) Band-pass filtering at 0.3-50 Hz, 3) Segmenting EEG signals into 1-second (200 timepoints), obtaining 3394 EEG segments in each session for each subject.

### E.2 MOTOR IMAGERY (MI)

We employ the BCIC-IV-2a as a representative dataset of MI classification tasks (Brunner et al., 2008). It contains EEG recordings in a four-class motor-imagery task from nine subjects with two repeated session each on different days. During the task, the subjects were instructed to imagine four types of movements (*i.e.*, right hand, left hand, feet, and tongue) for four seconds. Each session consists of a total of 288 trials with 72 trials for each type of the motor imagery. The EEG signals were recorded by 22 Ag/AgCl EEG electrodes in a sampling rate of 250 Hz. We segment the 22-channel EEG signals into 1-second segments, resulting in totally 1152 EEG data for each subject.

### E.3 EPILEPSY DETECTION (ED)

The CHB-MIT, one of the largest and most used public datasets for epilepsy, is adopted as a representative dataset of ED tasks (Shoeb & Guttag, 2010). It recorded 877.39 hours of multi-channel EEG in a sampling rate of 256 Hz from 23 pediatric patients with intractable seizures. However, as the montages (*i.e.*, the number and the places of electrodes) of EEG signals vary significantly among different subjects' recordings, we select to use only the EEG recordings with the same 23 channels (see Appendix A) and discard other channels or the recordings don't have all these 23 channels. Due to the purpose is to test whether the backdoor attack works on the ED task, not to study the epilepsy EEG classification, we segment part of the CHB-MIT dataset to form a four-class ED dataset (*i.e.*, the preictal, ictal, postictal, and interictal phases). Specifically, for a ictal phase EEG recording of $t_i$ seconds from $[s_i, e_i]$ timepoints, we segment the $[s_i - t_i, e_i]$ EEG as the preictal phase, the $[e_i, e_i + t_i]$ EEG as the postictal phase, and another $t_i$ seconds EEG recordings as the interictal phase which satisfying there is no ictal phase within half an hour before or after. Then we segment the 23-channel EEG signals into 1-second segments, consequently, there are 41336 segments left in total from all subjects, 10334 for each phase. As the imbalanced amount of data across different subjects, we separate these 41336 segments into 10 groups and treat the ten groups as 10 subjects.

## F IMPLEMENTATION DETAILS

### F.1 EXPERIMENT COMPUTING RESOURCES

We use two servers for conducting our experiments. A server with one Nvidia Tesla V100 GPU is used for running reinforcement learning, the CUDA version is 12.3. Another server with four Nvidia RTX 3090 GPUs is used for running the backdoor attacks, the CUDA version is 11.4.

### F.2 DETAILS OF BASELINE METHODS

In our Professor X backdoor attacks, for an EEG segment $x_i \in \mathbb{R}^{E \times T}$, we modify the $\beta F$ frequency-points and $\gamma E$ electrodes of a EEG segments with a constant number.

There are four baseline methods in our study for multi-target backdoor attacks, two of them are non-stealthy attacks (**PatchMT** and **PulseMT**) and two are stealthy attacks (**CompressMT** and **AdverseMT**). In order to achieve a fair comparison, we modify only first $\gamma E$ electrodes for all baseline attack methods. For the non-stealthy attacks, which are all on the temporal domains, we modify $\beta T$ timepoints of EEG signals. For the stealthy attacks, there is no constraint of the numbers of the modify timepoints as these attacks achieve stealthiness in another way.

For each baseline method, we try our best to find out the best performance, as demonstrated below. We promise that we did not maliciously lower the performances of the baseline methods.

### F.2.1 PATCHMT

PatchMT is a multi-trigger and MT extension of BadNets (Gu et al., 2019) where we fill the first $\beta T$ timepoints and $\gamma E$ electrodes of a EEG segments with a constant number. Specifically, for an EEG segment $x_i \in \mathbb{R}^{E \times T}$, we set the first $\gamma E$ electrodes and the first $\beta T$ timepoints of the EEG segment to a constant number. We normalize the EEG segment $x_i \in \mathbb{R}^{E \times T}$ to let $\mathbf{x}_i$'s mean is 0 and std is 1. Then set the first $\gamma E$ electrodes and the first $\beta T$ timepoints of $\mathbf{x}_i$ to a different constant number

for different class. The constant number for each class of $\{0, 1, 2, 3\}$ for four classes, and $\{-0.1, 0.0, 1.0\}$ for three classes. Finally, denormalize $\mathbf{x}_i$ to original signal $x_i$'s scale to generate $x_i^p$.

Although we try our best to find the best performance of PatchMT, and BadNets (Gu et al., 2019) is really efficient in image backdoor attacks, PatchMT cannot have satisfactory results in EEG BCI attack.

### F.2.2 PULSEMT

For PulseMT, we met the same questions as the PatchMT: how to identify the amplitude of each NPP signal for each class? If the numbers are too large then normal EEG signals, it will be unfair. If the numbers are too small, the efficacy of PulseMT is too negative.

We normalize the EEG segment $x_i \in \mathbb{R}^{E \times T}$ to let $\mathbf{x}_i$'s mean is 0 and std is 1. The constant amplitude for each class of $\{-0.8, -0.3, 0.3, 0.8\}$. Finally, denormalize $\mathbf{x}_i$ to original signal $x_i$'s scale to generate $x_i^p$.

### F.2.3 COMPRESSMT

Compressing the amplitude of EEG signals in the temporal domain will not change the morphology and the frequency distribution of EEG signals, thus obtaining stealthiness. For three-class Emotion datasets, the compress rate is $\{0.8, 0.6, 0.4\}$. For four-class Motor Imagery and Epilepsy datasets, the compress rate is $\{0.8, 0.6, 0.4, 0.2\}$.

### F.2.4 ADVERSEMT

AdverseMT is another stealthy EEG backdoor attacks, which is the multi-trigger and multi-target extension of adversarial spatial filter attacks (Meng et al., 2024), in wihch, for EEG segment $x_i \in \mathbb{R}^{E \times T}$, it learns an Spatial Filter $\mathbf{W} \in \mathbb{R}^{E \times E}$ by the adversarial loss to let the model $f$ misclassify $x_i$:

$$\min_{\mathbf{W}} \mathbb{E}_{(x_i, y_i) \sim \mathcal{D}}[-\mathcal{L}_{CE}(\mathbf{W}x_i, y_i) + \alpha \mathcal{L}_{MSE}(\mathbf{W}x_i, x_i)], \tag{10}$$

However, the original version of (Meng et al., 2024) requires the access to all training dataset $\mathcal{D}$ and the control of the training process of the model $f$. We modify the AdverseMT to only access to the training dataset $\mathcal{D}_{train}$. Note that the adversarial loss dose not have the special design for multi-target backdoor attacks, we only run the process $c$ times for obtaining $c$ spatial filters for different classes. So the poisoned subset are $\mathcal{S}_p = \{(\mathbf{W}_0(x), 0), (\mathbf{W}_1(x), 1), (\mathbf{W}_2(x), 2), (\mathbf{W}_3(x), 3)\}$.

### F.3 REINFORCEMENT LEARNING POLICY NETWORK ARCHITECTURE

Here, we design a concise but effective convolutional neural networks as the our policy network, which is defined as belows:

Table 8: The Architecture of Policy Network

| Layer | In | Out | Kernel | Stride |
|---|---|---|---|---|
| Conv2d | 1 | 32 | (1, 3) | (1, 1) |
| BatchNorm2d | | | | |
| ELU | | | | |
| AvgPool2d | | | | (1,2) |
| Conv2d | 32 | 64 | (1, 3) | (1, 1) |
| BatchNorm2d | | | | |
| ELU | | | | |
| AvgPool2d | | | | (1,2) |
| AdaptiveAvgPool2d | | | | (1, 1) |
| Flatten | | | | |
| Linear | 64 | 256 | | |

### F.4 TARGET EEG BCIS' NETWORK ARCHITECTURE

Three mostly-used EEG BCI models in real-world applications are investigated in our experiments, covering convolutional neural network (CNN) and recurrent neural network (RNN): 1) EEGNet (Lawhern et al., 2018), 2) DeepCNN (Schirrmeister et al., 2017), 3) LSTM (Tsiouris et al., 2018). Below we detail the architecture of each network. The EEGNet and DeepCNN are almost the same as the original paper (modified a little for cross-subject setting), LSTM comprises an embedding layer, a one-layer LSTM and a linear classifiers.

EEGNet is a compact and concise convolutional network for EEG BCI, having been proven to be effective in a variety of EEG fields with only 3 convolutional layers. DeepCNN is a little bit deeper than EEGNet, which comprises 4 blocks, 5 convolutional layers in total. The LSTM written by us, as demonstrated in Table 11, is a very shallow network. Our goal is to develop a model-agnostic BA method for EEG modality.

Table 9: The Architecture of EEGNet

| Layer | Kernel | Input Size | Output Size |
|---|---|---|---|
| $16 \times$ Conv1d | $(C, 1)$ | $C \times T$ | $16 \times 1 \times T$ |
| BatchNorm | | $16 \times 1 \times T$ | $16 \times 1 \times T$ |
| Transpose | | $16 \times 1 \times T$ | $1 \times 16 \times T$ |
| Dropout | 0.25 | $1 \times 16 \times T$ | $1 \times 16 \times T$ |
| $4 \times$ Conv2d | $(2 \times 32)$ | $1 \times 16 \times T$ | $4 \times 16 \times T$ |
| BatchNorm | | $4 \times 16 \times T$ | $4 \times 16 \times T$ |
| Maxpool2D | (2,4) | $4 \times 16 \times T$ | $4 \times 8 \times T/4$ |
| Dropout | 0.25 | $4 \times 8 \times T/4$ | $4 \times 8 \times T/4$ |
| $4 \times$ Conv2d | $(8 \times 4)$ | $4 \times 8 \times T/4$ | $4 \times 8 \times T/4$ |
| BatchNorm | | $4 \times 8 \times T/4$ | $4 \times 8 \times T/4$ |
| Maxpool2D | (2,4) | $4 \times 8 \times T/4$ | $4 \times 4 \times T/16$ |
| Dropout | 0.25 | $4 \times 4 \times T/16$ | $4 \times 4 \times T/16$ |
| Softmax Regression | | $4 \times 4 \times T/16$ | Class Number |

Table 10: The Architecture of DeepCNN

| Layer | Kernel | Input Size | Output Size |
|---|---|---|---|
| $F_1 \times$ Conv1d | $(1, 32)$ | $C \times T$ | $F_1 \times C \times T$ |
| BatchNorm | | $F_1 \times 1 \times T$ | $F_1 \times 1 \times T$ |
| $F_1 \times$ Conv1d | $(C, 1)$ | $F_1 \times C \times T$ | $F_1 \times 1 \times T$ |
| BatchNorm | | $F_1 \times 1 \times T$ | $F_1 \times 1 \times T$ |
| MaxPooling | (1,2) | $F_1 \times 1 \times T$ | $F_1 \times 1 \times T/2$ |
| Dropout | 0.25 | $F_1 \times 1 \times T/2$ | $F_1 \times 1 \times T/2$ |
| $F_2 \times$ Conv2d | $(1 \times 10)$ | $F_1 \times 1 \times T/2$ | $F_2 \times 1 \times T/2$ |
| BatchNorm | | $F_2 \times 1 \times T/2$ | $F_2 \times 1 \times T/2$ |
| Maxpool2D | (1,2) | $F_2 \times 1 \times T/2$ | $F_2 \times 1 \times T/4$ |
| Dropout | 0.25 | $F_2 \times 1 \times T/4$ | $F_2 \times 1 \times T/4$ |
| $F_3 \times$ Conv2d | $(1 \times 10)$ | $F_2 \times 1 \times T/4$ | $F_3 \times 1 \times T/4$ |
| BatchNorm | | $F_3 \times 1 \times T/4$ | $F_3 \times 1 \times T/4$ |
| Maxpool2D | (1,4) | $F_3 \times 1 \times T/4$ | $F_3 \times 1 \times T/16$ |
| Dropout | 0.25 | $F_3 \times 1 \times T/16$ | $F_3 \times 1 \times T/16$ |
| $F_4 \times$ Conv2d | $(1 \times 4)$ | $F_3 \times 1 \times T/16$ | $F_4 \times 1 \times T/16$ |
| BatchNorm | | $F_4 \times 1 \times T/16$ | $F_4 \times 1 \times T/16$ |
| Maxpool2D | (1,4) | $F_4 \times 1 \times T/16$ | $F_4 \times 1 \times T/64$ |
| Dropout | 0.25 | $F_4 \times 1 \times T/64$ | $F_4 \times 1 \times T/64$ |
| Softmax Regression | | $F_4 \times 1 \times T/64$ | Class Number |

Table 11: The Architecture of LSTM, $n$ is the embedding size.

| Layer | Input Size | Output Size |
|---|---|---|
| Linear ReLU | $C \times T$ | $n \times T$ |
| Linear | $n \times T$ | $n \times T$ |
| LSTM | $n \times T$ | $n \times T$ |
| Softmax Regression | $n \times T$ | Class Number |

# G    ATTACK PERFORMANCE OF PROFESSOR X

## G.1    DIFFERENT POISONING RATES

We present the performance of each backdoor attacks' performance under different poisoning rates in Table 12. We can see that our Professor X outperforms other baseline at all poisoning rates, demonstrating the superiority of Professor X. Note that the performance of Professor X on the MI dataset is significantly robust to low poisoning rates, i.e., ASR of 1.000 when $\rho = 0.05$.

## G.2    HYPERPARAMETER ANALYSIS: FREQUENCY AND ELECTRODES INJECTION RATIO

We present the performance of each backdoor attacks performance under different rates in Table 13 and Table 14. It can be observed with the increment of $\beta$ and $\gamma$, the attack performance increases. Because the trigger is bigger in clean EEG data.

## G.3    HYPERPARAMETER ANALYSIS IN REINFORCEMENT LEARNING

We applied the following reward function to acquire the optimal mask strategies for each triggers:

$$Q_t = \text{CA} + \lambda \, \text{ASR} + \mu \, \text{dis}(\mathcal{M}_f^{c_i}) + \nu \min(\mathcal{M}_f^{c_i}), \qquad (11)$$

where the first part means the clean accuracy, the second part means the attack success rate, the third part is aiming to scatter the injection positions in various frequency bands, and the fourth part is aiming to inject as high frequencies in EEG signals as possible. Here, we give a simple example to demonstrate the reward function. For an 10 timepoints long EEG segment $x_i$, $\widetilde{x}_i = \mathcal{F}(x_i)$. If the $\mathcal{M}_f^{c_i} = \{2, 3, 5, 7, 9\}$, because the minimal distance between each pair in $\mathcal{M}_f^{c_i}$ is $|2 - 3| = 1$, thus $\text{dis}(\mathcal{M}_f^{c_i}) = 1$. The $\min(\mathcal{M}_f^{c_i})$ means the lowest position in $\mathcal{M}_f^{c_i}$, thus $\min(\mathcal{M}_f^{c_i}) = 2$.

The analysis of the $\lambda$ are presented in Table 15. When $\lambda$ increase, the Attack performance increases while the Clean performance declines slightly.

Table 15: Clean (/C) and attack (/B) performance with ASR's hyperparameter $\lambda$, $\mu = 0.3, \nu = 0.005$

| | Dataset | Emotion | | Motor Imagery | | Epilepsy | |
|---|---|---|---|---|---|---|---|
| | Method | Clean | Attack | Clean | Attack | Clean | Attack |
| 0.5 | Professor X | $0.542_{\pm 0.03}$ | $0.847_{\pm 0.04}$ | $0.327_{\pm 0.02}$ | $1.000_{\pm 0.01}$ | $0.500_{\pm 0.04}$ | $0.922_{\pm 0.04}$ |
| 1.0 | Professor X | $0.537_{\pm 0.02}$ | $0.855_{\pm 0.03}$ | $0.325_{\pm 0.02}$ | $1.000_{\pm 0.01}$ | $0.482_{\pm 0.03}$ | $0.935_{\pm 0.05}$ |
| 2 | Professor X | $0.535_{\pm 0.03}$ | $0.857_{\pm 0.02}$ | $0.323_{\pm 0.02}$ | $1.000_{\pm 0.01}$ | $0.477_{\pm 0.04}$ | $0.944_{\pm 0.02}$ |

Table 12: Clean (/C) and attack (/B) performance with different poisoning rates for Professor X and other baseline methods. The target model is EEGNet for all cases.

| $\rho$ | Dataset | Emotion | | Motor Imagery | | Epilepsy | |
|---|---|---|---|---|---|---|---|
| | Method | Clean | Attack | Clean | Attack | Clean | Attack |
| 0.05 | PatchMT | 0.390 | 0.333 | 0.281 | 0.791 | 0.449 | 0.365 |
| | PulseMT | 0.488 | 0.337 | 0.275 | 0.788 | 0.473 | 0.397 |
| | ComprsMT | 0.448 | 0.313 | 0.269 | 0.754 | 0.449 | 0.329 |
| | Professor X | 0.491 | 0.566 | 0.321 | 1.000 | 0.460 | 0.667 |
| 0.10 | PatchMT | 0.443 | 0.334 | 0.279 | 0.785 | 0.452 | 0.400 |
| | PulseMT | 0.445 | 0.394 | 0.281 | 0.796 | 0.486 | 0.591 |
| | ComprsMT | 0.509 | 0.323 | 0.270 | 0.778 | 0.446 | 0.337 |
| | Professor X | 0.541 | 0.718 | 0.320 | 1.000 | 0.452 | 0.734 |
| 0.15 | PatchMT | 0.455 | 0.335 | 0.285 | 0.805 | 0.439 | 0.414 |
| | PulseMT | 0.438 | 0.514 | 0.280 | 0.787 | 0.447 | 0.669 |
| | ComprsMT | 0.488 | 0.332 | 0.275 | 0.792 | 0.461 | 0.374 |
| | Professor X | 0.528 | 0.805 | 0.322 | 1.000 | 0.460 | 0.781 |
| 0.20 | PatchMT | 0.481 | 0.334 | 0.277 | 0.816 | 0.461 | 0.451 |
| | PulseMT | 0.447 | 0.555 | 0.285 | 0.810 | 0.451 | 0.692 |
| | ComprsMT | 0.470 | 0.347 | 0.270 | 0.795 | 0.458 | 0.394 |
| | Professor X | 0.538 | 0.773 | 0.321 | 1.000 | 0.447 | 0.799 |
| 0.25 | PatchMT | 0.487 | 0.335 | 0.281 | 0.820 | 0.444 | 0.483 |
| | PulseMT | 0.466 | 0.701 | 0.275 | 0.815 | 0.431 | 0.684 |
| | ComprsMT | 0.493 | 0.335 | 0.269 | 0.800 | 0.462 | 0.427 |
| | Professor X | 0.551 | 0.836 | 0.325 | 1.000 | 0.447 | 0.834 |
| 0.30 | PatchMT | 0.459 | 0.343 | 0.280 | 0.809 | 0.440 | 0.496 |
| | PulseMT | 0.486 | 0.810 | 0.272 | 0.816 | 0.451 | 0.716 |
| | ComprsMT | 0.499 | 0.331 | 0.269 | 0.825 | 0.455 | 0.481 |
| | Professor X | 0.526 | 0.829 | 0.320 | 1.000 | 0.451 | 0.756 |
| 0.35 | PatchMT | 0.437 | 0.341 | 0.285 | 0.805 | 0.448 | 0.510 |
| | PulseMT | 0.437 | 0.767 | 0.275 | 0.837 | 0.482 | 0.757 |
| | ComprsMT | 0.473 | 0.347 | 0.265 | 0.851 | 0.446 | 0.517 |
| | Professor X | 0.489 | 0.763 | 0.321 | 1.000 | 0.453 | 0.910 |
| 0.40 | PatchMT | 0.490 | 0.345 | 0.283 | 0.824 | 0.460 | 0.549 |
| | PulseMT | 0.454 | 0.771 | 0.270 | 0.825 | 0.439 | 0.443 |
| | ComprsMT | 0.464 | 0.361 | 0.269 | 0.865 | 0.437 | 0.450 |
| | Professor X | 0.528 | 0.849 | 0.323 | 1.000 | 0.477 | 0.944 |

Table 13: Clean (/C) and attack (/B) performance with frequency injection rate $\beta$, $\gamma = 0.5$

| $\beta$ | Dataset | Emotion | | Motor Imagery | | Epilepsy | |
|---|---|---|---|---|---|---|---|
| | Method | Clean | Attack | Clean | Attack | Clean | Attack |
| 0.05 | PatchMT | 0.411 | 0.334 | 0.272 | 0.801 | 0.476 | 0.499 |
| | PulseMT | 0.464 | 0.752 | 0.265 | 0.800 | 0.505 | 0.670 |
| | Professor X | 0.522 | 0.744 | 0.319 | 0.999 | 0.482 | 0.923 |
| 0.10 | PatchMT | 0.431 | 0.363 | 0.283 | 0.824 | 0.482 | 0.540 |
| | PulseMT | 0.460 | 0.795 | 0.270 | 0.825 | 0.486 | 0.704 |
| | Professor X | 0.522 | 0.813 | 0.323 | 1.000 | 0.500 | 0.944 |
| 0.15 | PatchMT | 0.413 | 0.371 | 0.275 | 0.821 | 0.464 | 0.587 |
| | PulseMT | 0.449 | 0.701 | 0.271 | 0.821 | 0.477 | 0.632 |
| | Professor X | 0.532 | 0.848 | 0.322 | 0.998 | 0.477 | 0.947 |
| 0.20 | PatchMT | 0.390 | 0.377 | 0.271 | 0.829 | 0.479 | 0.644 |
| | PulseMT | 0.434 | 0.769 | 0.270 | 0.819 | 0.484 | 0.606 |
| | Professor X | 0.529 | 0.882 | 0.325 | 0.999 | 0.486 | 0.950 |
| 0.25 | PatchMT | 0.406 | 0.385 | 0.267 | 0.835 | 0.491 | 0.673 |
| | PulseMT | 0.491 | 0.705 | 0.275 | 0.832 | 0.478 | 0.566 |
| | Professor X | 0.519 | 0.865 | 0.328 | 0.999 | 0.486 | 0.941 |
| 0.30 | PatchMT | 0.417 | 0.382 | 0.269 | 0.831 | 0.464 | 0.706 |
| | PulseMT | 0.425 | 0.708 | 0.273 | 0.844 | 0.488 | 0.592 |
| | Professor X | 0.521 | 0.862 | 0.330 | 0.999 | 0.495 | 0.940 |
| 0.35 | PatchMT | 0.435 | 0.373 | 0.270 | 0.841 | 0.475 | 0.734 |
| | PulseMT | 0.423 | 0.621 | 0.276 | 0.839 | 0.479 | 0.589 |
| | Professor X | 0.527 | 0.850 | 0.332 | 0.998 | 0.496 | 0.947 |
| 0.40 | PatchMT | 0.438 | 0.378 | 0.271 | 0.843 | 0.469 | 0.751 |
| | PulseMT | 0.481 | 0.624 | 0.272 | 0.845 | 0.485 | 0.592 |
| | Professor X | 0.521 | 0.893 | 0.330 | 0.999 | 0.501 | 0.951 |
| 0.45 | PatchMT | 0.460 | 0.385 | 0.266 | 0.844 | 0.481 | 0.742 |
| | PulseMT | 0.429 | 0.633 | 0.277 | 0.856 | 0.499 | 0.601 |
| | Professor X | 0.519 | 0.877 | 0.325 | 0.999 | 0.492 | 0.962 |
| 0.50 | PatchMT | 0.423 | 0.386 | 0.263 | 0.840 | 0.480 | 0.752 |
| | PulseMT | 0.459 | 0.514 | 0.273 | 0.851 | 0.492 | 0.610 |
| | Professor X | 0.528 | 0.893 | 0.329 | 1.000 | 0.497 | 0.970 |

Table 14: Clean (/C) and attack (/B) performance with electrodes injection rate $\gamma$, $\beta = 0.1$

| $\gamma$ | Dataset | Emotion | | Motor Imagery | | Epilepsy | |
|---|---|---|---|---|---|---|---|
| | Method | Clean | Attack | Clean | Attack | Clean | Attack |
| 0.10 | PatchMT | 0.431 | 0.334 | 0.268 | 0.795 | 0.470 | 0.529 |
| | PulseMT | 0.425 | 0.498 | 0.269 | 0.802 | 0.502 | 0.717 |
| | ComprsMT | 0.407 | 0.349 | 0.271 | 0.805 | 0.482 | 0.656 |
| | Professor X | 0.489 | 0.485 | 0.235 | 0.367 | 0.499 | 0.814 |
| 0.20 | PatchMT | 0.473 | 0.335 | 0.271 | 0.805 | 0.464 | 0.599 |
| | PulseMT | 0.469 | 0.707 | 0.270 | 0.816 | 0.502 | 0.737 |
| | ComprsMT | 0.465 | 0.363 | 0.268 | 0.812 | 0.514 | 0.704 |
| | Professor X | 0.481 | 0.709 | 0.235 | 0.367 | 0.486 | 0.860 |
| 0.30 | PatchMT | 0.423 | 0.343 | 0.272 | 0.803 | 0.486 | 0.613 |
| | PulseMT | 0.488 | 0.767 | 0.273 | 0.814 | 0.506 | 0.749 |
| | ComprsMT | 0.451 | 0.398 | 0.271 | 0.811 | 0.494 | 0.700 |
| | Professor X | 0.500 | 0.743 | 0.235 | 0.367 | 0.490 | 0.883 |
| 0.40 | PatchMT | 0.453 | 0.343 | 0.270 | 0.812 | 0.478 | 0.525 |
| | PulseMT | 0.467 | 0.786 | 0.271 | 0.816 | 0.498 | 0.688 |
| | ComprsMT | 0.443 | 0.361 | 0.270 | 0.820 | 0.506 | 0.634 |
| | Professor X | 0.491 | 0.767 | 0.235 | 0.367 | 0.478 | 0.912 |
| 0.50 | PatchMT | 0.431 | 0.363 | 0.270 | 0.813 | 0.472 | 0.552 |
| | PulseMT | 0.460 | 0.795 | 0.269 | 0.819 | 0.471 | 0.710 |
| | ComprsMT | 0.430 | 0.366 | 0.269 | 0.821 | 0.503 | 0.640 |
| | Professor X | 0.522 | 0.813 | 0.235 | 0.367 | 0.477 | 0.944 |
| 0.60 | PatchMT | 0.452 | 0.377 | 0.267 | 0.819 | 0.480 | 0.549 |
| | PulseMT | 0.460 | 0.808 | 0.269 | 0.823 | 0.490 | 0.672 |
| | ComprsMT | 0.459 | 0.368 | 0.271 | 0.826 | 0.499 | 0.534 |
| | Professor X | 0.488 | 0.828 | 0.235 | 0.367 | 0.495 | 0.950 |
| 0.70 | PatchMT | 0.443 | 0.368 | 0.272 | 0.812 | 0.497 | 0.525 |
| | PulseMT | 0.437 | 0.809 | 0.270 | 0.821 | 0.459 | 0.716 |
| | ComprsMT | 0.456 | 0.366 | 0.273 | 0.835 | 0.492 | 0.571 |
| | Professor X | 0.527 | 0.853 | 0.235 | 0.367 | 0.489 | 0.955 |
| 0.80 | PatchMT | 0.461 | 0.383 | 0.268 | 0.821 | 0.479 | 0.573 |
| | PulseMT | 0.456 | 0.771 | 0.267 | 0.829 | 0.488 | 0.699 |
| | ComprsMT | 0.431 | 0.383 | 0.270 | 0.833 | 0.488 | 0.475 |
| | Professor X | 0.539 | 0.865 | 0.235 | 0.367 | 0.489 | 0.960 |
| 0.90 | PatchMT | 0.439 | 0.400 | 0.271 | 0.817 | 0.478 | 0.540 |
| | PulseMT | 0.461 | 0.811 | 0.269 | 0.823 | 0.494 | 0.694 |
| | ComprsMT | 0.459 | 0.389 | 0.274 | 0.836 | 0.490 | 0.309 |
| | Professor X | 0.520 | 0.824 | 0.235 | 0.367 | 0.489 | 0.970 |
| 1.00 | PatchMT | 0.430 | 0.370 | 0.267 | 0.823 | 0.476 | 0.526 |
| | PulseMT | 0.456 | 0.794 | 0.271 | 0.829 | 0.482 | 0.716 |
| | ComprsMT | 0.453 | 0.376 | 0.269 | 0.830 | 0.490 | 0.334 |
| | Professor X | 0.532 | 0.846 | 0.235 | 0.367 | 0.491 | 0.978 |

# H    MORE VISUALIZATION RESULTS

In this section, we plot the reconstructed triggers and masks on three datasets in Section H.1, then plot more visualizations of backdoor samples in Section H.2, and plot the learning curve of our reinforcement learning in Section H.3.

## H.1    NEURAL CLEANSE: RECONSTRUCTION TRIGGER PATTERNS

Here, we present more visualization in Figure 9, Figure 10, and Figure 11 of the reconstructed trigger patterns and mask patterns for each possible label on three dataset (*i.e.*, the CHB-MIT dataset, the BCIC-IV-2a dataset and the SEED dataset) the target model is EEGnet. It can be observed that the reconstructed trigger patterns and mask patterns of the clean models and Professor X backdoor-injected models are very similar to each other. Thus, our Professor X backdoor attack can easily bypass the defense of Neural Cleanse.

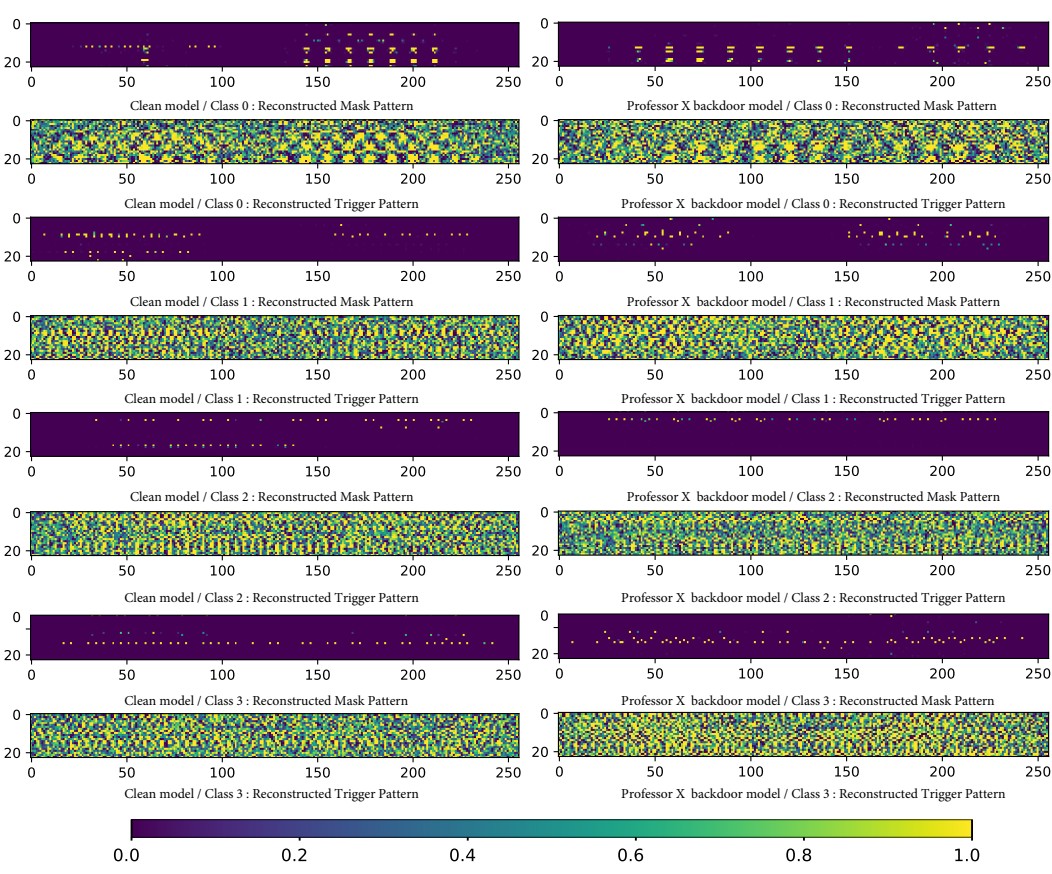

Figure 9: The reconstructed trigger patterns and mask patterns for each possible class in the CHB-MIT dataset. The results in the left column are reconstructed based on the clean model, the results in the right column are reconstructed based on the backdoor model. The EEG segments in the CHB-MIT dataset have 23 electrodes and 256 timepoints.

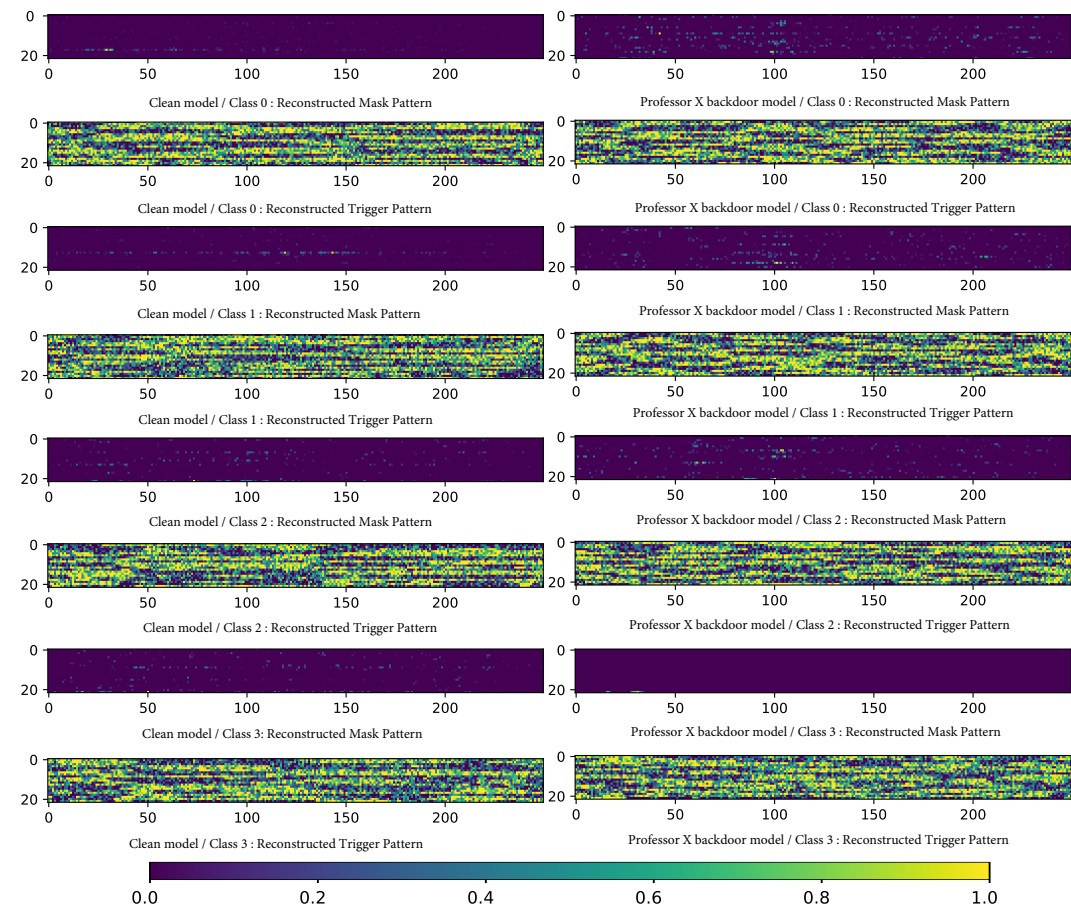

Figure 10: The reconstructed trigger patterns and mask patterns for each possible class in the MI dataset. The results in the left column are reconstructed based on the clean model, the results in the right column are reconstructed based on the backdoor model. The EEG segments in the MI dataset have 22 electrodes and 250 timepoints.

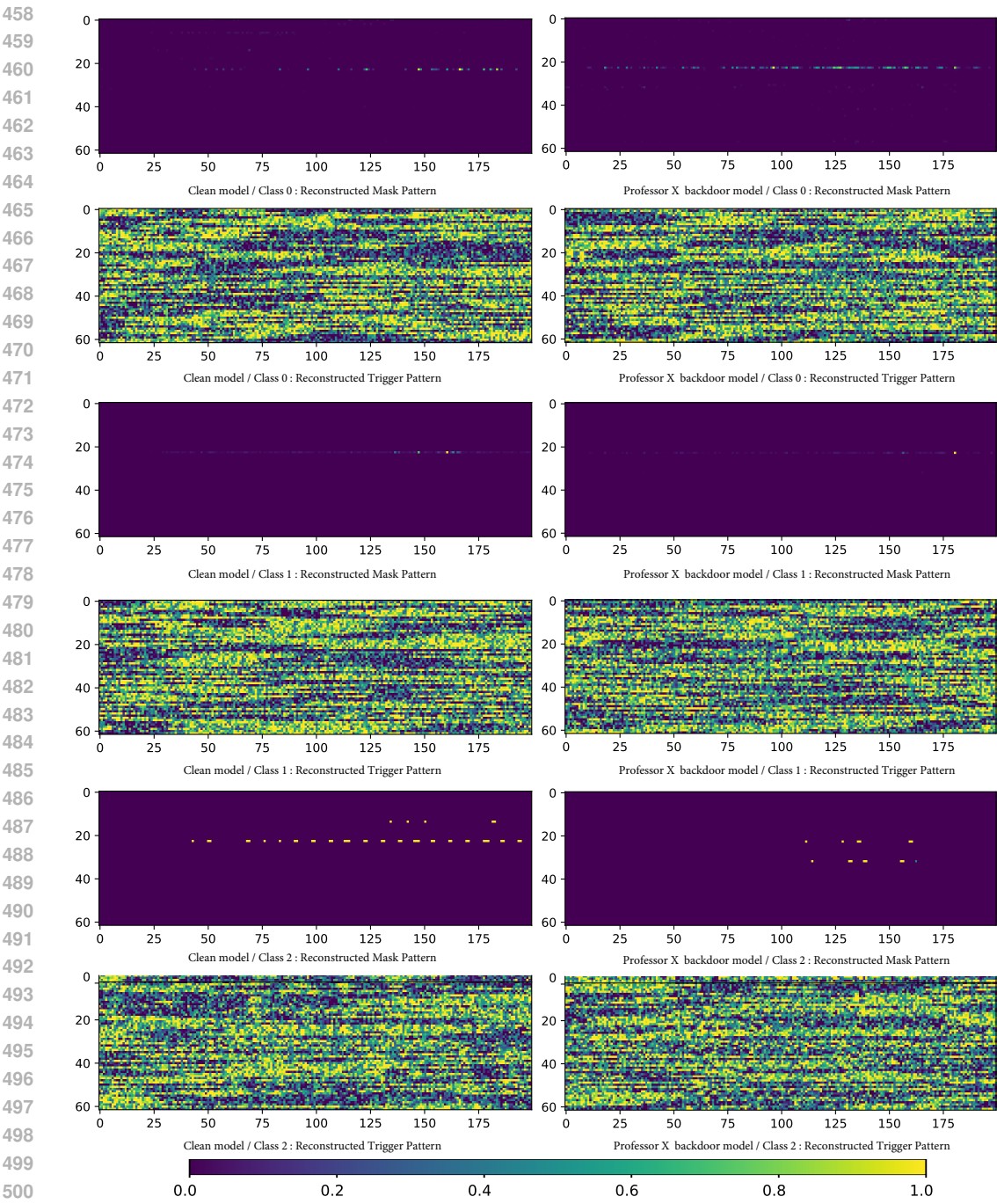

Figure 11: The reconstructed trigger patterns and mask patterns for each possible class in the ER dataset (i.e., SEED dataset). The results in the left column are reconstructed based on the clean model, the results in the right column are reconstructed based on the backdoor model. The EEG segments in the SEED dataset have 62 electrodes and 200 timepoints.

## H.2 VISUALIZATION OF BACKDOOR ATTACK SAMPLES

We present more visualization of the backdoor attack samples generated by our Professor X on three datasets in Fig 12, 13, and 14. The x-axis is the timepoints, the y-axis is the normalized amplitude. Top row: w.o. HF loss; Bottom row: with HF loss. Each column indicates each possible class.

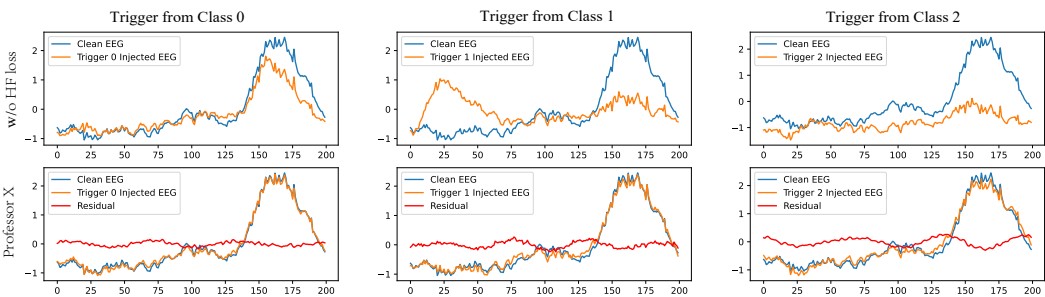

Figure 12: The Clean EEG (Blue), Trigger-injected EEG (Orange) and the Residual (Red) of the ER dataset.

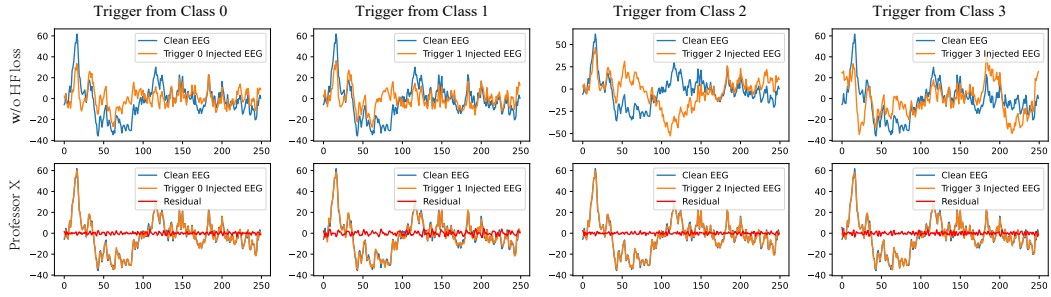

Figure 13: The Clean EEG (Blue), Trigger-injected EEG (Orange) and the Residual (Red) of the MI dataset.

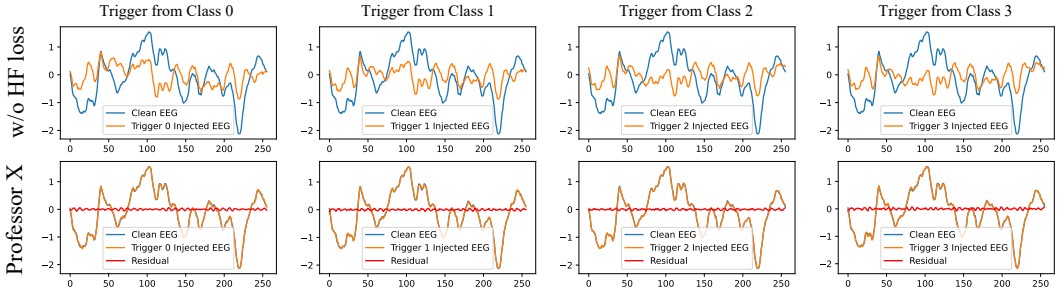

Figure 14: The Clean EEG (Blue), Trigger-injected EEG (Orange) and the Residual (Red) of the ED dataset.

### H.3 VISUALIZATION OF LEARNING CURVES OF REINFORCEMENT LEARNING

We present the visualization of the learning curves of the reinforcement learning of three dataset in Fig 15. We can see the effectiveness of our reinforcement, which converged within 50 epochs on the ER dataset, that is, only trained 50 backdoor models with different injection strategies. Our RL is more effective on the MI dataset and ED dataset, which finds a good strategy within less 10 epochs. Our RL is robust when learning strategies for different triggers as demonstrated in Fig 15(c) and (d), where the learning curves are quite similar when RL is performing on different triggers.

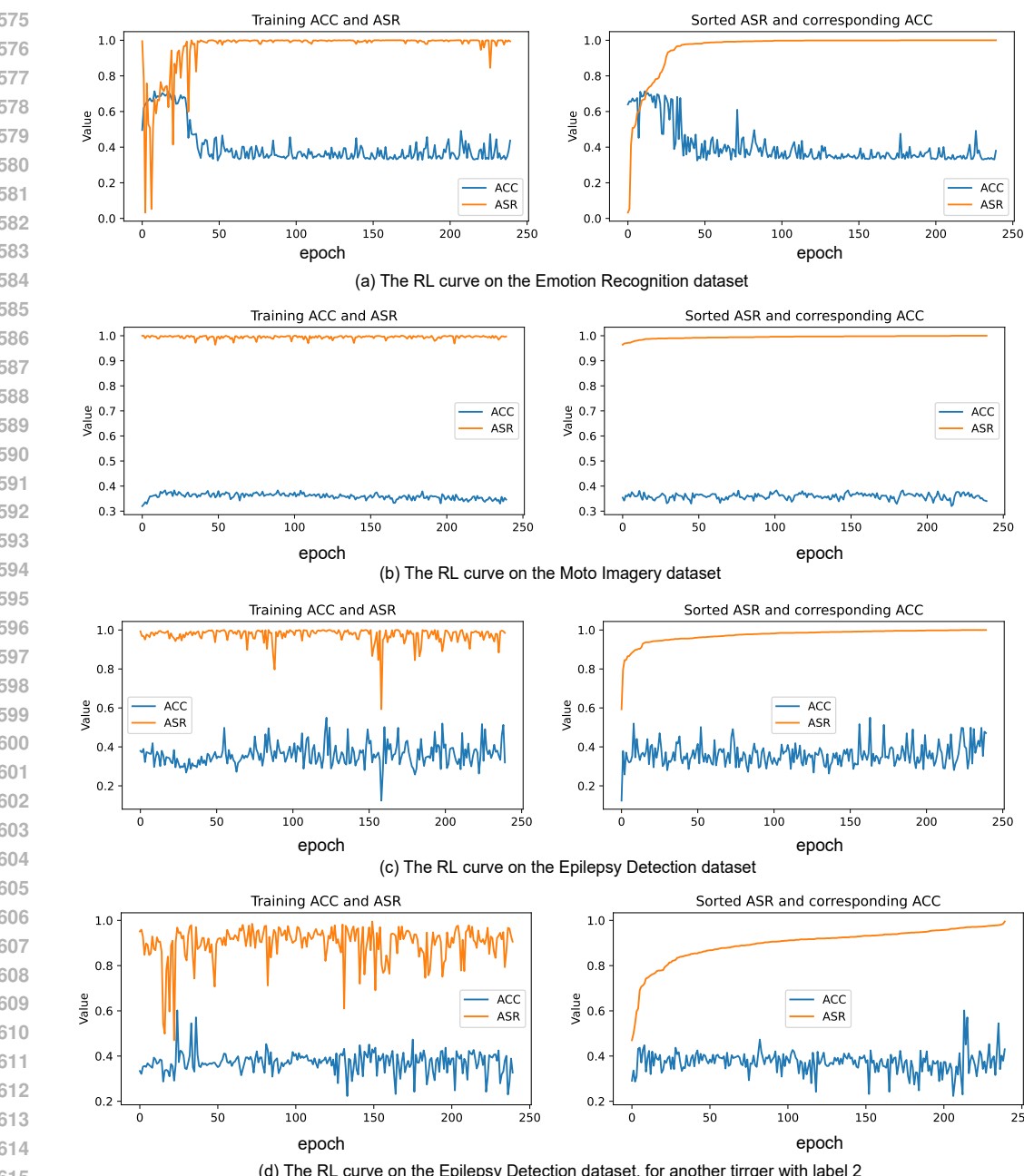

(a) The RL curve on the Emotion Recognition dataset

(b) The RL curve on the Moto Imagery dataset

(c) The RL curve on the Epilepsy Detection dataset

(d) The RL curve on the Epilepsy Detection dataset, for another tirrger with label 2

Figure 15: The learning curves of RL on three datasets. The right column is the curve we sort the (Clean Accuracy and Attack Success Rate) (ACC,ASR) according to the ASR. The backdoor models are all EEGNet.

