# OpenReview forum: "Professor X: Manipulating EEG BCI with Invisible and Robust Backdoor Attack"
_ICLR.cc/2025/Conference — Submitted to ICLR 2025_

### Official Review · Reviewer_dg1E · 2024-10-28

**Soundness:** 2
**Presentation:** 2
**Contribution:** 1
**Rating:** 5
**Confidence:** 5

**Summary:**

This paper presents Professor X, a novel backdoor attack method specifically designed for EEG-based brain-computer interfaces (BCIs). The proposed approach employs a three-stage clean-label poisoning strategy that includes trigger selection, reinforcement learning to identify optimal injection techniques, and the generation of poisoned data in the frequency domain.

**Strengths:**

The introduction of a three-stage clean-label poisoning strategy represents an innovative approach to backdoor attacks in the context of EEG-based BCIs.

The use of reinforcement learning to optimize injection techniques enhances the potential effectiveness of the attack and contributes to the growing body of research on adversarial techniques in neurotechnology.

**Weaknesses:**

The literature review is notably limited, overlooking key studies such as Liu et al. (2021) and Zhang & Wu (2019), which highlight the vulnerabilities of EEG-based BCIs to adversarial attacks on signal integrity.

This oversight significantly diminishes the impact and originality of the research, as the proposed method lacks validation against established vulnerabilities within existing literature.

A more thorough engagement with relevant studies would enhance the study's contributions and contextual relevance.

Reference:
Liu, Z., Meng, L., Zhang, X., Fang, W., & Wu, D. (2021). Universal adversarial perturbations for CNN classifiers in EEG-based BCIs. Journal of Neural Engineering, 18(4), 0460a4.
Zhang, X., & Wu, D. (2019). On the vulnerability of CNN classifiers in EEG-based BCIs. IEEE transactions on neural systems and rehabilitation engineering, 27(5), 814-825.

**Questions:**

What motivated the specific design choices made in the three-stage clean-label poisoning strategy?

How does the proposed method compare in effectiveness to existing backdoor attack methods in the literature?

---

> ### Author Response · Authors · 2024-11-19
> **Response to the reviewer dg1E (1/2)**
>
> Thanks for your time and efforts in reviewing our work! Your questions about the motivation of our attack's design is insightful, which also provide an opportunity to summarize our contributions. Below we carefully address them.Below we have addressed your questions and concerns point-by-point.
>
> ### Weaknesses
> > The literature review is notably limited ... A more thorough engagement with relevant studies would enhance the study's contributions and contextual relevance.
>
> Thanks for providing the additional references and we have added them in the revised paper, please kindly refer to the new version of our paper. However, we would like to kindly clarify that paper [1,2] are about adversarial attack on EEG BCI models, which is **not the same as backdoor attack**. In the revised paper, we have clarified the difference.
>
> Although backdoor attack and adversarial attack are all about the vulnerability of deep models, **backdoor attack is different from adversarial attack** in two ways: **(1) Attacking phase**: while adversarial perturbation attacks the model in the inference phase, backdoor attack injects a backdoor in the training phase; **(2) Attacking objective**: while adversarial attack aims to let deep models misclassify (the attacker doesn't care about the target class models will misclassify), backdoor attack aims to let deep models misclassify the samples with particular triggers to target class (the attacker clearly knows the target classes models will misclassify, thus can manipulate the model's output by injecting different triggers).
>
> Moreoever, the adversarial perturbation can also be used as a trigger for backdoor attack, which has been researched in [3]. In our paper, **we have compared [3] (the baseline AdverMT)** at the multi-trigger and multi-target settings in our paper. Please kindly refer to Table 1, it can be observed that our attack outperforms the adversarial-based backdoor attack. As the adversarial perturbation is designed for single-target attack, it fails to attack multi-target classes.
>
> In conclusion, we would like to kindly argue that the **overlook of two less relevant papers [1,2] won't diminishes the impact and originality of our research**. We sincerely hope you can reconsider your score.
>
> [1] X Zhang, et al. "On the vulnerability of CNN classifiers in EEG-based BCIs", IEEE TNSRE, 27(5):814–825, 2019.
>
> [2] Z Liu, et al. "Universal adversarial perturbations for CNN classifiers in EEG-based BCIs", 2021.
>
> [3] L. Meng, et al. "Adversarial filtering based evasion and backdoor attacks to EEG-based brain-computer interfaces", Information Fusion, 2024.

---

> ### Author Response · Authors · 2024-11-19
> **Response to the reviewer dg1E (2/2)**
>
> ### Questions
> > What motivated the specific design choices made in the three-stage clean-label poisoning strategy?
>
> Thanks for your interest in the design of Professor X! Actually, we design our attack in a reverse order, that is, we **(1)** firstly design the frequency injection stage, **（2）** then design the reinforcement learning stage, **(3)** lastly design the trigger selection from true dataset to do clean label attack.
> With this reverse order, you can better understand the motivation behind our design, and you will find the design of Professor X is very **intuitive**.
>
> 1. We firstly notice that all backdoor attacks for time-series and EEG BCI are injecting triggers in the temporal domain, which will inevitably bring unnatural frequency into the frequency domain. **Thus, we pioneerly propose to inject triggers in the frequency domain.** We combine two data's amplitude in the frequency domain to generate poisoned data. Since these two data are all natural data, the generated poisoned data is natural no matter in the temporal or frequency domain. We visualize the poisoned data and clean data in Fig 2 (frequency domain) and Fig 8 (temporal domain) to verify the stealthiness.
>
> 2. Now we know that injecting trigger in the frequency domain is stealthy and natural, but we don't know **which electrodes or frequencies should we inject the trigger**. Since for different EEG tasks, the EEG electrodes and frequencies strongly related to the performance of EEG BCI are different. **For the first time, we propose to adopt some optimization algorithm for leaning the injecting strategy.** After lots of experiments, we chose the reinforcement learning as our optimization algorithm. Moreover, we design two novel losses (DIS and HF) to enhance the stealthiness and robustness.
>
> 3. Last but not least, we have ensured the performance of Professor X, but we notice that certain classes of EEG data have specific morphology that can easily be identified by human experts, like the EEG during the ictal phase contains more spike/sharp waves than those during the normal state phase. In short, **different classes of EEG data have different frequency distribution**. To enhance the stealthiness, **we introduce the clean label attack**. The poisoned data has the same label as the trigger data, so the poisoned data's frequency information dosen't contain any frequency distribution from other classes.
>
> Consequently, we design our novel Professor X, a three-stage clean-label poisoning attack.
>
> > How does the proposed method compare in effectiveness to existing backdoor attack methods in the literature?
>
> We compare our attack with four baselines in our experiments, the results are displayed in Table 2 and Fig 3. We detail the experiment settings in section 4.3. Specifically,
>
> 1. We generate the poisoned data with different backdoor attack methods, and mixing these poisoned data into a bigger clean training dataset to form a poisoned dataset.
>
> 2. We train an EEG BCI model on the poisoned dataset, the trained EEG BCI model is injected with a backdoor, we call it backdoor model.
>
> 3. We test the backdoor model on the clean test set, and the poisoned test set where we inject triggers into the clean data. We report the clean accuracy and attack success rate by running this experiment.
>
> Except for the different methods used to generate poisoning data, the other experimental setups are all same. By this experiment we compare our attack with previous baselines.
>
> ---
> We hope this response could help address your concerns. We believe that this work contributes to this community and has the potential to serve as a catalyst for its development. We would sincerely appreciate it if you could reconsider your rating and we are more than happy to address any further concerns you may have. Thanks again!

---

> ### Author Response · Authors · 2024-11-25
> **Official Comment by Authors**
>
> Thank you for your thoughtful and insightful suggestions! We believe we have comprehensively addressed your questions regarding relative literature, Professor X's specific design, its effectiveness compared to other baselines including adversarial-based attacks.
> ﻿
>
> We would like to emphasize that our method is **the first to achieve both highly stealthy and robust backdoor attacks on EEG BCI**. Through data poisoning approach, our method even does not require controlling the training stage of target models. ﻿ ﻿
> ﻿
>
> We are wondering whether you have any additional questions or comments regarding our response to your review comments. We will do our best to address them. ﻿ ﻿
> ﻿
>
> We sincerely appreciate the time and effort you have dedicated to reviewing our manuscript. Thank you for your thoughtful consideration!

---

> ### Author Response · Authors · 2024-11-28
> **Kindly request for post-rebuttal comments**
>
> Dear Reviewer dg1E:
>
> Thanks for your valuable time in reviewing our paper! We are writing this comment to kindly request for post-rebuttal comments.
>
> **We have added the references you provided into out revised paper, and clarify the difference between our work and these references.** Have we already addressed your concern? Or do you have any further concerns?
>
> Please feel free to contact us, we are more than happy to hear from you. We would really appreciate if you could reconsider your rating and we are glad to address your further concerns.
>
> Best,
> Authors

---

> > ### Comment · Reviewer_dg1E · 2024-12-01
> >
> > Thank you for providing additional information and clarification to the concerns of myself and other reviewers. I have adjusted my score.

---

> > > ### Author Response · Authors · 2024-12-01
> > > **Thanks for the improved rating**
> > >
> > > Dear Reviewer dg1E:
> > >
> > > Thank you for the improved score! We are deeply grateful for your review, which has greatly assisted us in supplementing and perfecting our paper.
> > >
> > > Best,
> > > Authors

---

### Official Review · Reviewer_kEdv · 2024-10-29

**Soundness:** 4
**Presentation:** 4
**Contribution:** 4
**Rating:** 6
**Confidence:** 3

**Summary:**

This paper introduces "Professor X," a novel EEG backdoor attack designed to manipulate the outputs of electroencephalogram (EEG)-based brain-computer interfaces (BCIs). While EEG BCIs are commonly used for medical and device control applications, their safety has often been overlooked. Professor X addresses the limitations of existing EEG attacks, which typically focus on single-target classes and require interaction with the training stage of the BCI or lack stealthiness. This method uniquely considers the specific EEG electrodes and frequencies to be injected based on different EEG tasks and formats. Utilizing a reinforcement learning-based reward function enhances both robustness and stealthiness. Experimental results demonstrate Professor X's effectiveness, robustness, and generalizability, highlighting the potential vulnerabilities in EEG BCIs and urging the community to conduct defensive studies. Additionally, Professor X offers applications in protecting intellectual property within EEG datasets and BCI models by providing a concealed watermark. The attack exploits a three-stage clean label poisoning strategy: selecting triggers for each class, optimizing electrode and frequency injection strategies, and generating poisoned samples through spectral interpolation. Tests on various EEG task datasets confirm the method's efficacy and its ability to bypass existing defenses.

**Strengths:**

It introduces a novel method for manipulating EEG BCI outputs, filling a gap in the existing literature that largely focuses on single-target attacks.
The design incorporates reinforcement learning to enhance the attack's robustness and stealthiness, allowing it to evade detection more effectively than previous methods.
Professor X considers the specific EEG electrodes and frequency ranges relevant to different tasks and formats, making it adaptable to various EEG applications.
Experimental results demonstrate that Professor X is effective across multiple EEG tasks, indicating a broad applicability beyond a single context.

**Weaknesses:**

The potential for malicious use raises significant ethical issues, as manipulating EEG outputs could lead to harmful consequences for users and their applications.

The method involves a sophisticated three-stage clean label poisoning attack, which may be complex to implement in practice, especially for those lacking expertise in reinforcement learning and EEG signal processing.

The design's focus on particular EEG tasks might result in overfitting, reducing its effectiveness in more generalized scenarios or with novel tasks.

The approach may face challenges in scaling up for larger or more complex datasets, potentially limiting its effectiveness in real-world applications involving diverse user populations.

**Questions:**

What ethical frameworks are in place to govern the use of techniques like Professor X, and how can researchers ensure that such methods are not misused?

How effective is STRIP in detecting backdoor inputs under various levels of perturbation? Are there specific types of perturbations that significantly impact its performance?

How does STRIP compare to other defense mechanisms in terms of robustness against backdoor attacks like Professor X? What are its relative strengths and weaknesses?

 What specific mechanisms within the model lead to the observed drop in attack success rate (ASR) when Fine-Pruning is applied? Can these mechanisms be quantified?

How are low-activated neurons determined, and could this method inadvertently remove important features that are crucial for classification?

 How effective is Fine-Pruning at different pruning ratios beyond 0.7? Are there specific thresholds where the attack's effectiveness is significantly impacted?

What future research avenues could be explored to improve defenses against backdoor attacks like Professor X, particularly in the context of Fine-Pruning?

**Details Of Ethics Concerns:**

The potential for malicious use raises significant ethical issues, as manipulating EEG outputs could lead to harmful consequences for users and their applications.

---

> ### Author Response · Authors · 2024-11-19
> **Response to the reviewer kEdv (1/4)**
>
> We truly thank you for your appreciation of our work! Your question about how to defend Professor X is important and meaningful, which also helps us deepen the understanding of our Professor X attack. Below we have addressed your questions and concerns point-by-point.
>
> ### Weaknesses
>
> > The potential for malicious use raises significant ethical issues, as manipulating EEG outputs could lead to harmful consequences for users and their applications.
>
> We can not agree more! As we discussed in the general responses, the malicious use of our techniques raises severe ethical issuses. Unfortunately, we did not find any effective ways to defend Professor X due to the high stealthiness and robustness. We will try our best to find a method to guard our attack, and we here call for defensive research against Professor X.
>
> However, we would like to kindly argue that regarding the high stealthiness and robustness of our attack as a weakness is not suitable. While our work is aiming to alert the EEG community of the potential hazard of backdoor attack, addressing the ethical issues raised by our attack is somewhat out of our paper's scope.
>
> > The method involves a sophisticated three-stage clean label poisoning attack, which may be complex to implement in practice, especially for those lacking expertise in reinforcement learning and EEG signal processing.
>
> Thank you for rasing this concerns. We admit that our work is somewhat sophisticated, but every stage in our attack is meaningful and effective.
>
> 1. **We propose to inject triggers in the frequency domain,** which prevents from bring any unatrual frequency into the clean data.
>
> 2. **We propose to adopt some optimization algorithm for leaning the injecting strategy** to determine **which electrodes or frequencies should we inject the trigger**. Moreover, we design two novel losses (DIS and HF) to enhance the stealthiness and robustness.
>
> 3. **We introduce the clean label attack** to enhance the stealthiness since **different classes of EEG data have different frequency distribution**.
>
> You can see the clear motivation behind the design of these three stages, as each of them improves the stealthiness or effectiveness of Professor X. However, we can still simplify our attack in the following ways:
>
> 1. Our methods can be applied **without RL** if some performance drop is acceptable. As displayed in Table 3, removing the RL will cause about 15% drop in Attack performance.
>
> 2. If you do not have to consider the different frequency distribution across different classes, then **the trigger can be selected arbitrarily** without clean label attack.
>
> > The design's focus on particular EEG tasks might result in overfitting, reducing its effectiveness in more generalized scenarios or with novel tasks.
>
> We would like to take this opportunity to emphasize that we did not design our attack by focusing on any particular EEG tasks and our attack is generalizable across different EEG tasks, formats and EEG models. To verify the generalizability of our attack, we elaborately selected three EEG datasets as described in section 4.1. These datasets vary significantly in tasks, electrode numbers, montages, and sampling rates. The experimental results demonstrates the effectiveness across these different datasets, proving the generalizability.
>
> From table 1, our method has been proven to be effective when facing various situations (diverse EEG models, tasks, and formats). Furthermore, we evaluate our attack on another public dataset which studies the P300 tasks [1,2]. The attack performances of three different EEG models on the dataset are still excellent:
>
> | | Clean | ASR | 0 | 1 |
> |:-:|:-:|:-:|:-:|:-:|
> |EEGNet| 0.818 | 0.993 | 1.000 | 0.986 |
> |DeepCNN| 0.807 | 0.940 | 0.997 | 0.883 |
> |LSTM|  0.779 | 0.855 | 0.995 | 0.714 |
>
> It is worth mentioning that these results are obtained by only running the reinforcement learning 30 iterations, which takes only 0.5 hour on each model. These results can be another strong evidence to demonstrate the generalizability to other EEG datasets and real-world scenarios.
>
> So we are confident that our attack will be still effective in more generalized scenarios or with novel tasks. Or maybe we misunderstand what you meant, if you have any questions, please ask us.
>
> [1] U. Hoffmann, et al. "An efficient P300-based brain-computer interface for disabled subjects", J. Neurosci.Methods, 2008.
>
> [2] Rodrigo Ramele. P300-Dataset. https://www.kaggle.com/datasets/rramele/p300samplingdataset

---

> ### Author Response · Authors · 2024-11-19
> **Response to the reviewer kEdv (2/4)**
>
> > The approach may face challenges in scaling up for larger or more complex datasets, potentially limiting its effectiveness in real-world applications involving diverse user populations.
>
> We would like to clarify that our method is effective no matter how large the size of EEG datasets. As the we poison the dataset by a constant poisoning ratio, the larger the target dataset, the more poisoned data we inject into the dataset.
>
> Or the word "large" indicates the number of subjects in the dataset. However, our attack is still effecitve, as the experiments performed in our paper are all under cross-subject setting. We adopt the same experiment setup as that in the paper [3]. In short, we train the EEG model on the data from n-2 subjects, and test the EEG model on the data of a new subject. Our attack works very well under this cross-subject setting, so we can conclude that our method is effective no matter how large the number of subjects in the EEG datasets.
>
> As for the word "complex", we kindly guess that does this mean the complexity of classification task? Like, the number of label types. If so, we would like to argue that our attack is effective at most situations ( > 90%). Because in most cases, the number of label types in EEG BCI is no mare than four. The datasets selected in our experiment are the most complex datasets. For emotion recognition, most datasets only focus on binary classification, like DEAP [4] and DREAMER [5]. For motor imagery, most datasets only do binary classification. As far as we know, the BCIC-IV-2a is the only dataset to do four-class classification. For epilepsy detection, most datasets only do *itcal or non-itcal* binary classification. We have refined the task to four categories.
>
> From table 1 and the addtional experimental results on the P300 tasks above, we are quite confident that our attack will be effective facing challenges in scaling up for larger or more complex datasets.
>
> [3] Meng L, et al. EEG-based brain–computer interfaces are vulnerable to backdoor attacks[J]. IEEE Transactions on Neural Systems and Rehabilitation Engineering, 2023.
>
> [4] Koelstra S, et al. Deap: A database for emotion analysis; using physiological signals[J]. IEEE transactions on affective computing, 2011.
>
> [5] Katsigiannis S, Ramzan N. DREAMER: A database for emotion recognition through EEG and ECG signals from wireless low-cost off-the-shelf devices[J]. IEEE journal of biomedical and health informatics, 2017.
>
> ### Questions
>
> > What ethical frameworks are in place to govern the use of techniques like Professor X, and how can researchers ensure that such methods are not misused?
>
> We can require professionals in the field of EEG BCI to sign relevant guarantees that they will not use any harmful attack, but afterall this is not the fundamental solution. Only by developing an effective defensive or detection method can we stop the misuse of such techniques.
>
> Unfortunately, we did not find any effective ways to defend or detect Professor X due to the high stealthiness and robustness. To the best of our knowledge, we can not ensure that such methods won't be misused at this stage. We will do our utmost to find a way to protect our attacks. Here, we are calling for research on the defense against Professor X.
>
> > How effective is STRIP in detecting backdoor inputs under various levels of perturbation? Are there specific types of perturbations that significantly impact its performance?
>
> STRIP detects the backdoor based on thees findings: **1)** backdoor trigger is input-agnostic; **2)** backdoor trigger is strong and effective when performing input perturbation; **3)** The backdoor models' outputs (softmax) of poisoned data has very low entropy.
>
> As the original STRIP paper [6] said, their detection method is insensitive to the trigger-size, so STRIP is effective no matter the trigger is big or small. In our understanding of STRIP, it can detect the trigger that is obvious to models in the original space (like temporal domain for time series data, and spatial domain for image data).
>
> However, any trigger that may affect the above findings may cause STRIP's detection failure. For example, **1)** the trigger is input-specific; **2)** the trigger is not that strong, it fails when performing input perturbation; **3)** The trigger won't cause the backdoor model to predict with very low entropy.
>
> Our trigger is injected in the frequency domain, leading to the input-specific pattern in the temporal domain, causing **finding 1 to be invalid**. Moreover, the input perturbation in the temporal domain may damages the frequency information, causing our trigger disapper, leading to the **finding 2 to be invalid**. Last, as EEG is a nonstationary modality, the outputs of EEG models are always with high entropy, making **finding 3 to be invalid**. Thus, STRIP is not effective in detecting Professor X attack.
>
> We hope our answer addressed your concerns, if you have any questions, please feel free to ask.

---

> ### Author Response · Authors · 2024-11-19
> **Response to the reviewer kEdv (3/4)**
>
> > How does STRIP compare to other defense mechanisms in terms of robustness against backdoor attacks like Professor X? What are its relative strengths and weaknesses?
>
> Thanks for you interest in our work since you ask such an insightful question. We evaluated several backdoor defensive method in our paper, including Neural Cleanse [7], STRIP [6], Spectral Signature [8], and Fine-Pruning [9].
>
> STRIP has several strengths:
>
> **1) Insensitive to trigger-size**: Neural Cleanse tries to detect backdoor by reconstructing the trigger. However, Neural Cleanse is sensitive to the trigger-size, while STRIP is effective no matter the trigger is big or small. STRIP can detect the trigger that is obvious to models in the original space (like temporal domain for time series data, and spatial domain for image data).
>
> **2) Plug and Play**: Neural Cleanse needs the full control of the backdoor model as it inverse the trigger by adversarial loss. Spectral Signature needs the output of the last hidden layer, then perform singular value decomposition on the covariance matrix of these output. Fine-Pruning needs to detect the low-activated neurons and then prune them. While STRIP is plug and play, and compatible in any models. We only need the inputs and outpus of the backdoor models (treated as a blackbox as we don't need any intermediate outputs), then calculate the entropy of the outputs.
>
> **3) Backdoor model architecture-agnositc** STRIP only needs the inputs and outputs of the backdoor model, so it is an architecture-agnositc method and is generalize to many real-world application senarios.
>
> But as we discussed before, STRIP has some weaknesses:
>
> **1)** STRIP can only detect backdoor whose trigger is input-agnostic, it fails when the trigger is input-specific. While Neural Cleanse and Spectral Signature can guard some input-specific backdoor triggers.
>
> **2)** STRIP can only detect backdoor whose trigger is strong and still effective when performing input perturbation. It fails when the triggers disappear when performing input perturbation. While other backdoor defensive methods don't have this limitation.
>
> **3)** STRIP can only detect backdoor whose trigger causes models predict with low entropy. If the model's outputs are always of high entropy, STRIP fails. While Fine-Pruning doesn't have this limitation.
>
> > What specific mechanisms within the model lead to the observed drop in attack success rate (ASR) when Fine-Pruning is applied? Can these mechanisms be quantified?
>
> We are sorry that we don't have enough knowledge to perfectly answer this questions. The interpretation of EEG models is beyond our professional knowledge scope. But we try our best to share our understanding with you.
>
> As far as we know, Fine-Pruning [9] removes neurons in the model that are in a dormant state (i.e., have low activation values) when predicting clean samples, thereby disrupting backdoor behavior. So the drop in ASR is caused by pruning the neuron that is important to predict poisoned data (this neuron has high activation values when processing inputs with triggers). But we don't know whether these mechanisms can be quantified or not.
>
> > How are low-activated neurons determined, and could this method inadvertently remove important features that are crucial for classification?
>
> Fine-Pruning [9] assumes that the defender has a validation dataset $D_{valid}$ in which all data are clean. The defender feeds these clean data into the backdoor models, and recrods the average activation of each neuron. Afterwards, the defender iteratively prunes neurons from the DNN in increasing order of average activations. Thus, the low-activated neurons are those the average activation is low when feeding in clean data.
>
> Yes, Fine-Pruning might inadvertently remove important neurons that are crucial for classification. Because the average activation is obtained from the small subset $D_{valid}$, so the low-activated neurons determined by $D_{valid}$ may be high-activated neurons when feeding another clean validation dataset $D_{valid}’$. That is, the important neurons for classifying clean sample $x \in D_{valid}’$ may be low-activated neurons for all samples in $D_{valid}$, resulting in the pruning of these important neurons.

---

> ### Author Response · Authors · 2024-11-19
> **Response to the reviewer kEdv (4/4)**
>
> > How effective is Fine-Pruning at different pruning ratios beyond 0.7? Are there specific thresholds where the attack's effectiveness is significantly impacted?
>
> From Fig 7 in our paper, we can see that when pruning ratios are over 0.7, the attack success rates drop for all EEG tasks. However, the clean accuracies drop more greatly and approach the random level. So Fine-Pruning is not that effective for Professor X.
>
> In our understanding of Fine-Pruning, we would like to say that there is no specific thresholds where the attack's effectiveness is significantly impacted. Because the threshold in our paper seems to be 0.7, but this threshold is for EEGNet models. For other diverse EEG BCI models, many factors like the architecture, size, and activation function of the model will all have impact on the threshold.
>
> In practical applications of Fine-Pruning, the defender will set a threshold for clean accuracy, then prune the neurons until the clean accuracy drops to the threshold. So when the threshold is set relatively high, the backdoor may not be erased.
>
> > What future research avenues could be explored to improve defenses against backdoor attacks like Professor X, particularly in the context of Fine-Pruning?
>
> As we discussed above, Fine-Pruning requires that the defender has a validation dataset $D_{valid}$. The performance of Fine-Pruning relies heavily on the quality of the validation dataset, since the low-activated neurons are determined by the validation dataset.
>
> So in the future, building a large, diverse, high quality, and absolutely clean validation dataset is the key for improving the Fine-Pruning's performance. The most important part is the diversity, which not only means the diversity of EEG tasks, but also means the diversity of EEG formats. Thus, improving the defenses against backdoor attacks is not an easy task and needs joint efforts of the medical and academic communities.
>
> ---
> We hope our responses could help address your concerns. We believe that this work contributes to this community and has the potential to serve as a catalyst for its development. We would sincerely appreciate it if you could reconsider your rating and we are more than happy to address any further concerns you may have.
>
>
> [6] Gao Y, Xu C, Wang D, et al. Strip: A defence against trojan attacks on deep neural networks[C]//Proceedings of the 35th annual computer security applications conference. 2019: 113-125.
>
> [7] Bolun Wang, Yuanshun Yao, et al. Neural Cleanse: Identifying and mitigating backdoor attacks in neural networks. In 2019 IEEE Symposium on Security and Privacy (S&P), pp. 707–723. IEEE, 2019.
>
> [8] Brandon Tran, Jerry Li, and Aleksander Madry. Spectral signatures in backdoor attacks. Advances in Neural Information Processing Systems (NeurIPS), 31, 2018.
>
> [9] Kang Liu, Brendan Dolan-Gavitt, and Siddharth Garg. Fine-pruning: Defending against backdooring attacks on deep neural networks. In International Symposium on Research in Attacks, Intrusions, and Defenses, pp. 273–294. Springer, 2018.

---

> ### Author Response · Authors · 2024-11-25
> **Official Comment by Authors**
>
> Thank you for your thoughtful and insightful suggestions! We believe we have comprehensively addressed your questions regarding Professor X's sophisticated design, its generalizability on novel tasks, its scalability on larger or more complex datasets, and defensive research against Professor X.
> ﻿
>
> We would like to emphasize that our method is **the first to achieve both highly stealthy and robust backdoor attacks on EEG BCI**. Through data poisoning approach, our method even does not require controlling the training stage of target models. ﻿
> ﻿
>
> We are wondering whether you have any additional questions or comments regarding our response to your review comments. We will do our best to address them. ﻿
> ﻿
>
> We sincerely appreciate the time and effort you have dedicated to reviewing our manuscript. Thank you for your thoughtful consideration!

---

> ### Author Response · Authors · 2024-11-28
> **Kindly request for post-rebuttal comments**
>
> Dear Reviewer kEdv:
>
> Thanks for your recognition of our work and insightful questions about how to defend Professor X. We are writing this comment to kindly request for post-rebuttal comments.
>
> **We given point-by-point answer to each of your questions and we believe we have comprehensively addressed them. Plus, we have added new experiments and a new section on the defensive research in our revised paper.** Have we already addressed your concern? Or do you have any further concerns?
>
> Please feel free to contact us, we are more than happy to hear from you. We would really appreciate if you could reconsider your rating and we are glad to address your further concerns.
>
> Best,
> Authors

---

> ### Author Response · Authors · 2024-12-02
> **Reminder for Reviewer kEdv**
>
> Dear Reviewer kEdv:
>
> This is a gentle reminder that the discussion period ends in about 30 hours. We are pleased to report that after engaging in the discussion, other reviewers have raised their scores: Reviewer JuBu raised 1 point and Reviewer dg1E raised 2 points. We responded to your original review on 11/19 and wanted to check in to see if you have any further questions or comments. If you find the proposed revisions and the discussion here helpful in clarifying the paper and/or increasing its value, we kindly request that you comment to that effect and consider raising your score before the deadline. Please let us know if you have any final comments, as we aim to address any of your concerns by tomorrow, 12/2. Again, the final deadline for your response is in 30 hours. Thank you for your time and thoughtful consideration!
>
> Best,
> Authors

---

### Official Review · Reviewer_uEuL · 2024-10-31

**Soundness:** 4
**Presentation:** 4
**Contribution:** 3
**Rating:** 8
**Confidence:** 4

**Summary:**

This paper presents "Professor X," a new EEG backdoor attack aimed at influencing the outputs of electroencephalogram (EEG)-based brain-computer interfaces (BCIs). While EEG BCIs are widely utilized in medical and device control settings, their security has often been neglected. Professor X improves upon existing EEG attack methods, which typically target single classes and either require interaction with the BCI's training phase or lack stealth. This innovative approach strategically selects specific EEG electrodes and frequencies for injection based on various EEG tasks and formats. By employing a reinforcement learning-based reward function, the method enhances both robustness and stealth. Experimental results demonstrate Professor X's effectiveness, resilience, and generalizability, underscoring vulnerabilities in EEG BCIs and calling for further defensive research in the field. Additionally, Professor X can help protect intellectual property within EEG datasets and BCI models by embedding a concealed watermark. The attack employs a three-stage clean label poisoning strategy: selecting triggers for each class, optimizing injection strategies for electrodes and frequencies, and generating poisoned samples via spectral interpolation. Testing on diverse EEG task datasets validates the method’s effectiveness and its capacity to circumvent existing defenses.

**Strengths:**

It presents an innovative approach for manipulating EEG BCI outputs, addressing a gap in the existing literature that predominantly emphasizes single-target attacks. The design leverages reinforcement learning to improve the attack's robustness and stealth, enabling it to evade detection more successfully than earlier methods. Professor X takes into account the specific EEG electrodes and frequency ranges associated with different tasks and formats, making it versatile for various EEG applications. Experimental results show that Professor X is effective across multiple EEG tasks, highlighting its broad applicability beyond just one context.

**Weaknesses:**

The method might encounter difficulties when scaling to larger or more intricate datasets, which could restrict its effectiveness in real-world applications with varied user populations.

**Questions:**

What potential research directions could be pursued to enhance defenses against backdoor attacks such as Professor X, especially regarding Fine-Pruning techniques?

---

> ### Author Response · Authors · 2024-11-19
> **Response to the reviewer uEuL (1/2)**
>
> We truly thank you for your appreciation of our work! Your questions about the generalizability of our attack is insightful, which also provides an opportunity to summarize our contributions.  Our point-by-point responses are as follows.
>
> ### Weaknesses
> > The method might encounter difficulties when scaling to larger or more intricate datasets, which could restrict its effectiveness in real-world applications with varied user populations.
>
> We would like to clarify that our method is effective no matter how large  the size of EEG datasets. As the we poison the dataset by a constant poisoning ratio, the larger the target dataset, the more poisoned data we inject into the dataset.
>
> Or we may misunderstand the word "large", we kindly guess that does this mean the number of subjects in the dataset is large? If so, our attack is still effecitve, as the experiments performed in our paper are all under cross-subject setting. We adopt the same experiment setup as that in the paper [1]. In short, we train the EEG model on the data from n-2 subjects, and test the EEG model on the data of a new subject. Our attack works very well under this cross-subject setting, so we can conclude that our method is effective no matter how large the number of subjects in the EEG datasets.
>
> As for the word "intricate", we kindly guess that does this mean the intricacy of classification task? Like, the number of label types. If so, we would like to argue that our attack is effective at most situations ( > 90%). Because in most cases, the number of label types in EEG BCI is no mare than four. The datasets selected in our experiment are the most intricate datasets. For emotion recognition, most datasets only focus on binary classification, like DEAP [2] and DREAMER [3]. For motor imagery, most datasets only do binary classification. As far as we know, the BCIC-IV-2a is the only dataset to do four-class classification. For epilepsy detection, most datasets only do *itcal or non-itcal* binary classification. We have refined the task to four categories.
>
> From table 1, our method has been proven to be effective when facing various situations (diverse EEG models, tasks, and formats). Furthermore, we evaluate our attack on another public dataset which studies the P300 tasks [4,5]. The attack performances of three different EEG models on the dataset are still excellent:
>
> | | Clean | ASR | 0 | 1 |
> |:-:|:-:|:-:|:-:|:-:|
> |EEGNet| 0.818 | 0.993 | 1.000 | 0.986 |
> |DeepCNN| 0.807 | 0.940 | 0.997 | 0.883 |
> |LSTM|  0.779 | 0.855 | 0.995 | 0.714 |
>
> It is worth mentioning that these results are obtained by only running the reinforcement learning 30 iterations, which takes only 0.5 hour on each model. These results can be another strong evidence to demonstrate the generalizability to other EEG datasets and real-world scenarios.
>
> In conclusion, we are quite confident that our attack will be effective when encountering difficulties when scaling to larger or more intricate datasets.
>
> ---
> [1] Meng L, Jiang X, Huang J, et al. EEG-based brain–computer interfaces are vulnerable to backdoor attacks[J]. IEEE Transactions on Neural Systems and Rehabilitation Engineering, 2023, 31: 2224-2234.
>
> [2] Koelstra S, Muhl C, Soleymani M, et al. Deap: A database for emotion analysis; using physiological signals[J]. IEEE transactions on affective computing, 2011, 3(1): 18-31.
>
> [3] Katsigiannis S, Ramzan N. DREAMER: A database for emotion recognition through EEG and ECG signals from wireless low-cost off-the-shelf devices[J]. IEEE journal of biomedical and health informatics, 2017, 22(1): 98-107.
>
> [4] U. Hoffmann, et al. "An efficient P300-based brain-computer interface for disabled subjects", J. Neurosci.Methods, 2008.
>
> [5] Rodrigo Ramele. P300-Dataset. https://www.kaggle.com/datasets/rramele/p300samplingdataset

---

> ### Author Response · Authors · 2024-11-19
> **Response to the reviewer uEuL (2/2)**
>
> ### Questions
> > What potential research directions could be pursued to enhance defenses against backdoor attacks such as Professor X, especially regarding Fine-Pruning techniques?
>
> Thanks for your interest of defensive research against Professor X and we are also concerned about its potential dangers. We have evaluated five defensive methods to try to guard Professor X, but all of them failed. The Fine-Pruning method seems to success at the pruning ratio over 0.7, but it damages the clean task's performance.
>
> Fine-Pruning [4] assumes that the defender has a validation dataset $D_{valid}$ in which all data are clean. The defender feeds these clean data into the backdoor models, and records the average activation of each neuron. Afterwards, the defender iteratively prunes neurons from the DNN in increasing order of average activations. In practical applications of Fine-Pruning, the defender will set a threshold for clean accuracy, then prune the neurons until the clean accuracy drops to the threshold. So when the threshold is set relatively high, the backdoor may not be erased.
>
> As we discussed above, Fine-Pruning requires that the defender has a validation dataset $D_{valid}$. The performance of Fine-Pruning relies heavily on the quality of the validation dataset, since the low-activated neurons are determined by the validation dataset.
>
> So in the future, building a large, diverse, high quality, and absolutely clean validation dataset is the key for improving the Fine-Pruning's performance. The most important part is the diversity, which not only means the diversity of EEG tasks, but also means the diversity of EEG formats. Thus, improving the defenses against backdoor attacks is not an easy task and needs joint efforts of the medical and academic communities.
>
> ---
> We hope this response could help address your concerns. We believe that this work contributes to this community and has the potential to serve as a catalyst for its development. We are more than happy to address any further concerns you may have. Thanks again!
>
> [4] Kang Liu, Brendan Dolan-Gavitt, and Siddharth Garg. Fine-pruning: Defending against backdooring attacks on deep neural networks. In International Symposium on Research in Attacks, Intrusions, and Defenses, pp. 273–294. Springer, 2018.

---

> ### Author Response · Authors · 2024-11-25
> **Official Comment by Authors**
>
> Thank you for your thoughtful and insightful suggestions! We believe we have comprehensively addressed your questions regarding Professor X's generalizability on novel tasks, its scalability on larger or more complex datasets, and defensive research against Professor X. ﻿
>
> We would like to emphasize that our method is **the first to achieve both highly stealthy and robust backdoor attacks on EEG BCI**. Through data poisoning approach, our method even does not require controlling the training stage of target models. ﻿ ﻿
>
> We are wondering whether you have any additional questions or comments regarding our response to your review comments. We will do our best to address them. ﻿ ﻿
>
> We sincerely appreciate the time and effort you have dedicated to reviewing our manuscript. Thank you for your thoughtful consideration!

---

### Official Review · Reviewer_JuBu · 2024-11-04

**Soundness:** 2
**Presentation:** 2
**Contribution:** 2
**Rating:** 6
**Confidence:** 3

**Summary:**

The paper introduces Professor X, a novel, frequency-based EEG attack designed to be stealthy and multi-target. The method involves three main steps: 1) Finding triggers for each class, 2) using reinforcement learning to find optimal electrodes and frequencies injection
strategies, and 3) generating poisoned samples using triggers and clean data.

**Strengths:**

1. The study has a clear research question, which makes its purpose easy to understand.
3. The idea of employing reinforcement learning for finding the optimal electrodes and frequencies for data poisoning is interesting.
2. The authors designed multiple experiments to evaluate different parts of their method, and the ones focused on showing the method's robustness are especially valuable.

**Weaknesses:**

As mentioned in the related works, a research direction exists that focuses on designing frequency-based backdoor attacks. Although existing methods are designed for images rather than time series, the authors could still compare their proposed method with existing approaches to better highlight the novelty of the work in relation to current frequency-based methods.

The authors designed several baselines (stealthy and non-stealthy) based on BadNets, PP-based BD attacks, and so on, which is great. However, it would be great if the authors considered and designed some baselines based on the existing frequency-based BD attack, if applicable.

The stealthiness of the method is one of the claims of the method. Although there are some visualizations in this regard, it would be great if the authors designed an experiment to validate the stealthiness of the method. It may be similar to a previous study [1], which used anomaly detection methods.

The author only considers three models for the classifiers: EEGNet, DeepCNN, and LSTM. However, it would be great if the author considered other new models, like TIMESNET [2] and other new transformer-based models.

The quality of the writing needs improvement; here are some points:
The third paragraph of the introduction requires revision for clarity and coherence.
Figure 1 consists of five sub-figures that provide a good summary of the method. However, in the introduction (line 050), the authors begin by explaining Figure 1-d, which destroys the flow.

The Methodology section should be improved by first defining the key concepts, symbols, and problems.
It would also be helpful to include a table of abbreviations and symbols, as the multiple terms used throughout the paper may be confusing for readers.



[1]. Lin X, Liu Z, Fu D, Qiu R, Tong H. BACKTIME: Backdoor Attacks on Multivariate Time Series Forecasting. arXiv e-prints. 2024 Oct:arXiv-2410.

[2]. Wu H, Hu T, Liu Y, Zhou H, Wang J, Long M. TimesNet: Temporal 2D-Variation Modeling for General Time Series Analysis. InThe Eleventh International Conference on Learning Representations.

**Questions:**

In addition to the points in the 'Weaknesses' section, I'd like to add one more:

How effective is the proposed BD attack when applied to approaches that utilize frequency information of data, such as [3]?



[3] Zhang X, Zhao Z, Tsiligkaridis T, Zitnik M. Self-supervised contrastive pre-training for time series via time-frequency consistency. Advances in Neural Information Processing Systems. 2022 Dec 6;35:3988-4003.

---

> ### Author Response · Authors · 2024-11-19
> **Response to the reviewer JuBu (1/2)**
>
> Thanks for your valuable comments, which helps us to greatly improve our paper's quality! The additional experiments you requested have enriched our experimental results and better demonstrated the generalization and robustness of our method! We'd like to express our appreciation that our novel reinforcement learning and comprehensive experiments are well recognized. Below we carefully address your questions and concerns point-by-point.
>
> ### Weaknesses
> > However, it would be great if the authors considered and designed some baselines based on the existing frequency-based BD attack (FreBA in the following text), if applicable.
>
> Thanks for your careful reading! Actually **we have compared our model with the vanilla frequency-based BD attack**. First we would like to illustrate the difference between our attack and previous FreBA. Then your concern about the comparison of FreBA can be naturally resolved.
>
> Compared to the previous frequency-based backdoor attack, our attack has three differences: **(1) Multi-target vs Single-target:** our attack can fully control the output of classifier model, but previous FreBA can only attack one target class. **(2) Stealthiness of Time-Series Modality:** previous FreBA are designed for image modality, which will lose stealthiness while directly performing on the time-series modality, our attack introduce a novel HF loss to address it, showing in Fig 8. **(3) Reinforcement Learning:** we introduce RL to optimize the injecting strategy, which greatly improves the performance, stealthiness and robustness. Previous FreBA inject the trigger in a constant place.
>
> In the Table 3, we conducted an ablation study to verify the effectiveness of RL, please kindly refer to the revised paper. The variant **Random is basically a vanilla frequency-based BD attack.** The results showed our attack is better than the vanilla FreBA.
>
> > It would be great if the authors designed an experiment to validate the stealthiness of the method. It may be similar to a previous study [1], which used anomaly detection methods.
>
> Thanks for your valuable advice! We have added this experiment in our revised paper, in Table 6. The ROC-AUC is around 0.5 and F1-score is either around 0.5 or near 0 across all datasets, indicating that the detection results are nearly random guess. These strongly demonstrates the stealthiness of Professor X.
>
> ||ER F1-score|ER AUC|MI F1-score|MI AUC|ED F1-score|ED AUC
> |:-:|:-:|:-:|:-:|:-:|:-:|:-:
> |GDN|0.50|0.51|0.50|0.51|0.50|0.50
> |USAD|0.00|0.51|0.00|0.51|0.00|0.50
>
> > The author only considers three models for the classifiers: EEGNet, DeepCNN, and LSTM. However, it would be great if the author considered other new models, like TimesNet [1] and other new transformer-based models.
>
> Thanks for your constructive suggestions. We have added new experiment results in our revised paper, in Table 2. It can be seen that our attack works well (the ASRs are almost over 95%, except conformer on the ER task) on the TimesNet [1] and a new Transformer-based model EEG-conformer [2] (is cited over 200 times), further proving the generalizability of our attack. We have detailed the individual ASRs for each category in Table 1 of the main text, please kindly refer to the revised paper.
>
> It takes a lot of time to conduct the experiment with TimesNet, as this model is not designed for multi-channel data like EEG BCI, thus is inefficient in processing EEG signals. We use TimesNet as a singal-channel feature extrator and concatenate all features of all EEG channels. Then we feed the concatenated feature into a linear layer for EEG classification. Our code of TimesNet and EEG-conformer will be public too.
>
> ||ER Clean|ER Attack|MI Clean|MI Attack|ED Clean|ED Attack
> |:-:|:-:|:-:|:-:|:-:|:-:|:-:
> |TimesNet|0.485|0.956|0.276|0.997|0.373|0.986
> |EEG-conformer|0.475|0.894|0.296|0.996|0.419|0.944
>
> By the way, we would like to discuss the reasons why we chose EEGNet, DeepCNN and LSTM. These three models are the most widely used model in the EEG community. And for most real-world application of EEG BCI, only shallow and simple models are required because deep and sophisticated models may overfit when the data is limited. Thus, considering the real-world situation, we chose these three simple models, but it doesn't mean that our attack is not effective on new sophisticated models.
>
> We would like to take this opportunity to emphasize that our method is an EEG task-agnostic, model architecture-agnostic, and format-agnostic attack, which can generalize to many scenarios.
>
> ---
> [1] Wu H, Hu T, Liu Y, Zhou H, Wang J, Long M. TimesNet: Temporal 2D-Variation Modeling for General Time Series Analysis. In The Eleventh International Conference on Learning Representations.
>
> [2] Yonghao Song, Qingqing Zheng, Bingchuan Liu, and Xiaorong Gao. EEG conformer: Convolutional transformer for EEG decoding and visualization. IEEE Transactions on Neural Systems and Rehabilitation Engineering, 31:710–719, 2022.

---

> ### Author Response · Authors · 2024-11-19
> **Response to the reviewer JuBu (2/2)**
>
> > The quality of the writing needs improvement; here are some points: The third paragraph of the introduction requires revision for clarity and coherence. Figure 1 consists of five sub-figures that provide a good summary of the method. However, in the introduction (line 050), the authors begin by explaining Figure 1-d, which destroys the flow.
>
> Your suggestions are so helpful, thank you very much! We have reorder the subfigure in Figure 1 and refine our writing in the revised paper. Please kindly refer to the new version of our paper, where the modifications are marked in blue.
>
> > The Methodology section should be improved by first defining the key concepts, symbols, and problems. It would also be helpful to include a table of abbreviations and symbols, as the multiple terms used throughout the paper may be confusing for readers.
>
> What a great suggestion! We added a table of key symbols in the Appendix A (Table 7) due to the page limitation. And we remind the reader the symbol table at the beginning of methodology section. Please kindly refer to the new version of our paper.
>
> ### Questions
> > How effective is the proposed BD attack when applied to approaches that utilize frequency information of data, such as [3]?
>
> We are very sure that our attack is effective on any supervised learning-based models no matter they utilize frequency information or not. Because the TimesNet model contains a module to convert 1D time series data into structured 2D tensors using the frequency information, and our attack works well on TimesNet. So our attack is effective when applied to approaches that utilize frequency information of data.
>
> However, the paper [3] is a self-supervised contrastive learning method, which is not supervised learning. Even in the image field, BD attacking self-supervised learning is far different from BD attacking supervised learning [4], not to mention BD attacking contrastive learning [5]. Thus, we don't know whether our attack can work on [3], but we would like to kindly remind that the effectiveness of BD attack on [3] is out of our paper's scope.
>
> ---
> We hope our responses could help address your concerns. We believe that this work contributes to this community and has the potential to serve as a catalyst for its development. We would sincerely appreciate it if you could reconsider your rating and we are more than happy to address any further concerns you may have. Thanks again!
>
> [3] Zhang X, Zhao Z, Tsiligkaridis T, Zitnik M. Self-supervised contrastive pre-training for time series via time-frequency consistency. Advances in Neural Information Processing Systems. 2022 Dec 6;35:3988-4003.
>
> [4] Jia J, Liu Y, Gong N Z. Badencoder: Backdoor attacks to pre-trained encoders in self-supervised learning[C]//2022 IEEE Symposium on Security and Privacy (S&P). IEEE, 2022: 2043-2059.
>
> [5] Carlini N, Terzis A. Poisoning and Backdooring Contrastive Learning[C]//International Conference on Learning Representations. 2022, oral.

---

> ### Author Response · Authors · 2024-11-25
> **Official Comment by Authors**
>
> Thank you for your thoughtful and insightful suggestions! We believe we have comprehensively addressed your questions regarding Professor X's comparison with vanilla FreBA, its stealthiness against anomaly detection methods, its generalizability on new sophisticated models, and writing improvement. We added the additional experimental results and refined our writing in our revised submission.
> ﻿
>
> We would like to emphasize that our method is the **first to achieve both highly stealthy and robust backdoor attacks on EEG BCI**. Through data poisoning approach, our method even does not require controlling the training stage of target models.
> ﻿
>
> We are wondering whether you have any additional questions or comments regarding our response to your review comments. We will do our best to address them.
> ﻿
>
> We sincerely appreciate the time and effort you have dedicated to reviewing our manuscript. Thank you for your thoughtful consideration!

---

> > ### Comment · Reviewer_JuBu · 2024-11-26
> >
> > Thank you to the authors for their clear and detailed responses to my and the other reviewers' questions. This has improved my view of the paper, and I have raised my score to 6.

---

> > > ### Author Response · Authors · 2024-11-26
> > > **Official Comment by Authors**
> > >
> > > Thank you for your recognition of our paper and the improved score! We are deeply grateful for your review, which has greatly assisted us in supplementing and perfecting our paper.

---

### Official Review · Reviewer_RejD · 2024-11-06

**Soundness:** 1
**Presentation:** 1
**Contribution:** 1
**Rating:** 1
**Confidence:** 5

**Summary:**

From the outset, the abstract of the submission presents a proposition that appears to be unrealistic and somewhat disconnected from contemporary research realities. It remains inaccurate to assert that "While electroencephalogram (EEG) based brain-computer interfaces (BCIs) have been extensively employed in medical diagnosis, healthcare, and device control, the safety of EEG BCIs has long been neglected."

The manuscript constructs a fictitious framework, suggesting that research regarding BCIs has already translated into widespread applications. The submission relies on historical datasets such as BCIC-IV-2a. It is critical to note that motor imagery may not be effective for the target demographic of individuals with paralysis due to significant neural degeneration. Moreover, the methodology seems to rely on a dubious SEED database to fabricate artificial backdoor attack scenarios, ultimately suggesting solutions that are not based in rigorous academic research. This strategy does not promote the growth of academic inquiry, thus justifying a rejection of this submission.

**Strengths:**

The reviewer found no substantial strengths in the submission. It is fundamentally inadequate to fabricate problems only to propose solutions. Backdoor attacks do not present a significant concern in the field of BCI research at this time, as established paradigms are still lacking. Nonetheless, BCI research is experiencing significant growth due to advancements in machine learning; however, a considerable distance remains before it can transition to healthcare and broader applications that would necessitate the implementation of protections against backdoor attacks.

**Weaknesses:**

The prevailing conditions within the submission exhibit a lack of realism, an insufficiency of novel contributions, and a substantial deficiency in advancements for the BCI community.

**Questions:**

Why did the authors construct an entirely unrealistic and artificial scenario? Where did the authors encounter such hyperbolic or enthusiastic claims regarding the purported applications of BCI in healthcare and medical diagnostics? Currently, only a limited number of conditionally approved, mostly invasive devices have been tested on a small cohort of subjects within closed clinical studies.

---

> ### Author Response · Authors · 2024-11-19
> **Response to the reviewer RejD**
>
> We thank the reviewer for your time and effort in reviewing our work. Below we would like to exchange our ideas with you point-by-point.
>
> > It remains inaccurate to assert that "While electroencephalogram (EEG) based brain-computer interfaces (BCIs) have been extensively employed in medical diagnosis, healthcare, and device control,..."
>
> We would like to say that EEG BCIs have been widely employed is not wrong. Because the employment in academia is also employment. As you agree with that **BCI research is experiencing significant growth due to advancements in machine learning**. According to the Web of Science, the number of papers on EEG BCI has increased from 6,185 in 2010 to 16,470 in 2023.
>
> Our attack can be used for **falsifying EEG datasets**, which is significantly severe in academia. As one can **draw any neuroscience conclusion from a fake EEG dataset**. The negative impact of academic fraud goes without saying, especially in the field of life sciences.
>
> > The submission relies on historical datasets such as BCIC-IV-2a. Moreover, the methodology seems to rely on a dubious SEED database.
>
> We adopt three public dataset with high citation to ensure our experiments are credible. These datasets are covering three widely-studied EEG tasks: **1)** epilepsy detection (CHB-MIT), **2)** emotion recognition (SEED), **3)** and motor-imagery (BCIC-IV-2a).
>
> Since you questioned the BCIC-IV-2a and SEED dataset, we would like to know your opinion about the epilepsy detection CHB-MIT dataset. What do you think about the CHB-MIT dataset? Our attack is also effective on the CHB-MIT dataset, is this result unrealistic too?
>
> We sincerely hope that you could explain why you evaluate the SEED dataset **"dubious"**, and what factors do you refer to when you say **"dubious"**? According to the SEED Dataset's Website [1]: *As of December 2023, the cumulative number of applications and research institutions using SEED have reached more than 5800 and 1000, respectively.* Moreover, The SEED's paper has been cited by more than 1,900 papers. Are these 1900 papers dubious too?
>
> > It is critical to note that motor imagery may not be effective for the target demographic of individuals with paralysis due to significant neural degeneration.
>
> We would like to say that this fact doesn't mean the useless of motor imagery EEG BCI. According to your logic, we can say that **It is critical to note that GLASSES may not be effective for the target demographic of individuals with MYOPIA due to significant neural degeneration.** However, for other people with myopia due to other reasons such as long screen time, the glasses is very helpful. Hence, the fact you presented has nothing to do with the effectiveness of motor imagery EEG BCI.
>
> In contrast, for the many patients with paralysis due to other reasons like spinal cord injuries, the motor imagery BCI is very helpful. The EEG BCI already helps a paralytic walk again [2], so the wider application of EEG BCI is predictable.
>
> To be honest, we did not understand the rationality of your logic, why you present this fact and what do you want to illustrate by this fact? We sincerely hope you could explain it.
>
> ### Questions
>
> > Why did the authors construct an entirely unrealistic and artificial scenario?
>
> No, we didn't. Please kindly refer to the general response for more details.
>
> > Where did the authors encounter such hyperbolic or enthusiastic claims regarding the purported applications of BCI in healthcare and medical diagnostics?
>
> We, as EEG BCI researchers, are conducting cutting-edge EEG BCI research with multiple top hospitals to try to improve the life quality of epilepsy patients. Specifically, we are trying to provide early warning for epileptic seizures with the help of EEG BCI. We are quite sure that the research of epilepsy detection is meaningful.
>
> As the impressive results presented in [2], where a paralytic walk again with the help of EEG BCI, the motor imagery EEG BCI has been the hope of many paralytics.
>
> > Currently, only a limited number of conditionally approved, mostly invasive devices have been tested on a small cohort of subjects within closed clinical studies.
>
> 1. We would like to argue that we don't have to wait until the horse has bolted before close the stable door.
>
> 2. Although there are a limited number of subjects are taking the treat of BCI, we would like to say we still need to consider the security risk of the using of EEG BCI on them.
>
> 3. Please note that our attack can also be misused in academic researches for academic fraud.
>
> ---
>
> We hope this response could help address your concerns. We believe that this work contributes to this community and has the potential to serve as a catalyst for its development. We are more than happy to further exchange ideas with you.
>
> [1] https://bcmi.sjtu.edu.cn/home/seed
>
> [2] Lorach H, Galvez A, Spagnolo V, et al. Walking naturally after spinal cord injury using a brain–spine interface[J]. Nature, 2023, 618(7963): 126-133.

---

> ### Author Response · Authors · 2024-11-25
> **Official Comment by Authors**
>
> Thank you for your thoughtful and insightful suggestions. We believe we have comprehensively addressed your questions regarding Professor X's realism.
>
>
> We are wondering whether you have any additional questions or comments regarding our response to your review comments. We will do our best to address them. ﻿
>
>
> We sincerely appreciate the time and effort you have dedicated to reviewing our manuscript. Thank you for your thoughtful consideration!

---

> ### Author Response · Authors · 2024-11-27
> **Replying to the Reviewer RejD**
>
> Dear Review RejD:
>
> Thank you for your feedback and the valuable time in reviewing our paper! We would really appreciate if you can reply some of our response's questions. Since your latest feedback (not public to everyone and ACs) overlooked these questions. We list them below for saving your valuable time:
>
> 1. Why do you regard the SEED dataset dubious?
> 2. What's your opinion about the CHB-MIT dataset, are our attack's results on this dataset unrealistic too?
> 3. What do you want to illustrate by this fact "*It is critical to note that motor imagery may not be effective for the target demographic of individuals with paralysis due to significant neural degeneration*"?
>
> We sincerely hope to hear from you and discuss with these questions. This is important for us to better address your concerns.
>
> Best,
> Authors

---

> ### Comment · Reviewer_RejD · 2024-11-27
> **Response to author question:**
>
> A. Why do you regard the SEED dataset dubious?
>
> Answer: The SEED database, while popular, has methodological limitations. The long-duration film clips, despite single-emotion labeling, likely induced multiple affective states, compromising the accuracy of self-reported emotions. The laboratory setting may not fully capture natural emotional expression, and the focus on basic emotions limits its applicability to complex emotional states. Cultural biases, inherent in using a limited dataset, further complicate the interpretation of results. Additionally, the use of extended film clips to induce specific emotions for BCI applications may not be practical or effective.
>
> B. What's your opinion about the CHB-MIT dataset, are our attack's results on this dataset unrealistic too?
>
> Answer: The proposed scenario of a malicious actor remotely hacking a medical device to induce seizures in patients is highly unrealistic and impractical. Medical devices, especially those involved in life-critical functions, are typically isolated from public networks for security and regulatory compliance reasons. Cyberattacks on such devices almost always require physical access or sophisticated social engineering techniques.
> A more realistic and relevant research direction would be to focus on improving the security of medical device networks and developing robust defense mechanisms against potential threats. This could involve research on advanced firewall technologies, intrusion detection systems, and secure firmware updates.
>
> C. What do you want to illustrate by this fact "It is critical to note that motor imagery may not be effective for the target demographic of individuals with paralysis due to significant neural degeneration"?
>
> Answer: Sensorimotor Rhythm (SMR)-based motor imagery BCI is widely recognized as unsuitable for late-stage ALS (CLIS) patients due to cognitive decline and altered brain activity patterns [1,2]. While effective in earlier stages, when numerous non-BCI alternatives exist, alternative BCI approaches like P300-based or fNIRS-based systems are more appropriate for CLIS patients. While recent research has questioned certain aspects of [2], this seminal work remains a cornerstone in the BCI field. Therefore, exploring attacks on a paradigm with limited practical application in late-stage ALS (CLIS) is not an appropriate submission for a top-tier conference such as ICLR 2025.
>
> 1. Foerster BR, Welsh RC, Feldman EL. 25 years of neuroimaging in amyotrophic lateral sclerosis. Nat Rev Neurol. 2013 Sep;9(9):513-24. doi: 10.1038/nrneurol.2013.153. Epub 2013 Aug 6. PMID: 23917850; PMCID: PMC4182931.
>
> 2. Kübler A, Birbaumer N. Brain–computer interfaces and communication in paralysis: Extinction of goal directed thinking in completely paralysed patients?. Clinical neurophysiology. 2008 Nov 1;119(11):2658-66.

---

> ### Author Response · Authors · 2024-11-27
> **Replying to the Reviewer RejD**
>
> We greatly appreciate your timely response and valuable comments on the datasets! We are now more likely to know what your real concern is, and would like to kindly conclude it as: *The paradigm and applications of EEG BCI are not mature enough at the current stage, thus there is no need to focus on the safety issue*.
>
> ### Major point
> We would like to take this opportunity to emphasize the motivation of our paper, which will in some degree answer your concern:
> 1. Although the BCI applications is not mature enough at the current stage, **it is still necessary to pay attention to backdoors like Professor X for future BCI applications**. **We don't have to wait until the horse has bolted before close the stable door.** It is worth noting that EEG BCI is gradually moving towards practical application [1]. We are confident that BCI will have better practical applications in the future.
>
> 2. Our work provides a fundamental security assessment perspective for BCI society, promoting the real practical application of BCI. Since the primary consideration before real application is **security issues**.
>
> 3. EEG BCI is widely used in academia, our attack can be used for **falsifying EEG datasets**. We provided an example in the **General Response** to illustrate the harm of this malicious use, which we absolutely do not encourage. We hope that our research can serve as a catalyst in the BCI community, raising more attention to BCI safety and promoting its practical application.
>
> ### Minor point
> By the way, as you said P300-based method is more effective for CLIS patients, we also validated our attack on a P300 dataset [2] and found our attack is still effective. But after all the CLIS patients do not represent everyone who needs a motor imagery BCI, there are many other patients like those suffering from spinal cord injuries need motor imagery BCI. We admit motor imagery BCI is not effective for everybody, but as long as one patient benefits from it [1], it is useful.
>
> | | Clean | ASR | 0 | 1 |
> |:-:|:-:|:-:|:-:|:-:|
> |EEGNet| 0.818 | 0.993 | 1.000 | 0.986 |
> |DeepCNN| 0.807 | 0.940 | 0.997 | 0.883 |
> |LSTM|  0.779 | 0.855 | 0.995 | 0.714 |
>
> ---
> In conclusion, our work is not aiming to attack BCI that has already been applied in the real-world (*while Professor X has the ability*), but more about **alerting a severe potential hazard** of backdoor in EEG BCI and **calling for defensive research** (*that's why we added a new section in appendix to discuss how to defend backdoor like Professor X in EEG BCI*).
>
> The datasets used in our study is for validating our attack's efficacy across various EEG tasks and formats. There might be some drawbacks in these datasets, but after all it is not the focus of our work. Our work promotes the future practical application of EEG BCI (in the safety way) and the construction of a more honest academic community (in the academic integrity way).
>
> Thanks for your valuable time in discussing with us! We are more than happy to address any further concerns you may have.
>
> [1] Lorach H, Galvez A, Spagnolo V, et al. Walking naturally after spinal cord injury using a brain–spine interface[J]. Nature, 2023, 618(7963): 126-133.
> [2] Rodrigo Ramele. P300-Dataset. https://www.kaggle.com/datasets/rramele/p300samplingdataset

---

> > ### Comment · Reviewer_RejD · 2024-11-27
> > **It is not about dataset but complete misunderstanding of the BCI research field**
> >
> > Dear Authors,
> >
> > Thank you for your detailed response. However, the fundamental issue with your work, as accurately identified by the initial review ("From the outset, the abstract of the submission presents a proposition that appears to be unrealistic and somewhat disconnected from contemporary research realities."), remains its disconnect from real-world medical device security realities.
> >
> > The medical device field, particularly regarding pacemakers, DBS, cochlear implants, hearing aids, and future BCIs, is significantly more mature than AI or deep learning. This maturity has led to robust security protocols and regulatory oversight, such as those enforced by the FDA, EMA, etc. To date, there have been no documented instances of successful hacking or backdoor attacks on these devices. Medical devices are typically customized for individual patients, and their associated data and models are subject to stringent confidentiality requirements mandated by medical regulatory standards. User model customization for each session, necessitated by electrode placement variations, remains a significant challenge in EEG-based BCI. This practical issue warrants further research attention rather than theoretical explorations of unrealistic backdoor attacks.
> >
> > It is essential to conduct thorough research and familiarize oneself with the field before proposing unrealistic scenarios and potential solutions. While publicly available datasets can be valuable for research, it's crucial to recognize that future approved medical devices are subject to stringent regulations and security measures, limiting the potential for malicious exploitation.
> >
> > We urge you to consider applying your research efforts to fields such as image processing or speech recognition, where the regulatory landscape may be less restrictive.
> >
> > As a member of the BCI community, the reviewer cannot endorse a publication that disregards the established security practices and potential consequences of such claims. Therefore, the initial recommendation stands.

---

> ### Author Response · Authors · 2024-11-27
> **Replying to the Reviewer RejD**
>
> Dear Reviewer RejD:
>
> Thanks for your timely and detailed response! We are really glad that we and you have already reached some consensus. Below we tried to address your concerns:
>
> > Unrealistic problem.
>
> 1. We definitely agree with that the EEG BCI is not mature at the current stage, including the paradigm and applications. But the potential practical application of it cannot be denied (We believe you are agree with this as you are a member of the BCI community too). Before real application of EEG BCI, we must consider the security issues of it. Our work promotes developing more safer BCI for future real application, which is definitely not unrealistic.
>
> > it's crucial to recognize that future approved medical devices are subject to stringent regulations and security measures, limiting the potential for malicious exploitation.
>
> 2. We believe we and you have reached the consensus that **the primary consideration before real application is security issues**. Unfortunately, the security issues are definitely not limited by simple and vague "*stringent regulations and security measures*". First of all, how to draft these regulations? **If all members in the BCI community do not realize that BCI can be injected backdoor, there is no one to draft the regulations to regulate backdoor attacks in BCI**. Our work provides an important insight about the severe backdoor attacks threats to EEG BCI. However, we believe that there must be other security issues in BCI but are never been noticed. It would be dangerous if someone maliciously used some techniques unknown to the public. Our work promotes the draft of regulations for Backdoor Attack in BCI.
>
> > The medical device field, particularly regarding pacemakers, DBS,  ... This maturity has led to robust security protocols and regulatory oversight, such as those enforced by the FDA, EMA, etc.
>
> 3. The medical devices you cited including pacemakers, DBS, cochlear implants, etc, are not using the AI or deep learning (DL). So they definitely have not been affected by backdoor attacks aiming at AI models. These devices are developed based on the traditional electronic methods and the industry has adequate knowledge about these devices along with the potential hazard they may face, that's why robust security protocols and regulatory oversight can be made by the FDA, EMA, etc.
> We hope to reached the consensus with you that AI or DL has shown some superiority in BCI, due to the complexity and human-unreadbility of brain signals. In recent years, there has been an increasing number of articles developing EEG BCI with AI or DL methods. **However, the BCI community dose not have sufficient knowledge about AI or DL models along with the potential hazard they may face**. Our work is aiming to study and point out the potential hazard, promoting designing more safer BCI for future real application.
>
>
> > User model customization for each session, necessitated by electrode placement variations, remains a significant challenge in EEG-based BCI. This practical issue warrants further research attention rather than theoretical explorations of unrealistic backdoor attacks... We urge you to consider applying your research efforts to fields such as image processing or speech recognition...
>
> 4. Actually, we are BCI researchers, not safety researchers in the image processing or speech recognition fields. We definitely agree with that the studies for user model customization and electrode placement, etc, are essential in BCI. However, studying security issues is equally important, and these two are not in conflict.  **Just like the traditional mature medical devices, there are some researchers focus on improving their performance, while there must be some researchers focus on improving their safety.**
> We noticed the security issues of EEG BCI, although we know the whole BCI society is focusing on improving the decoding accuracy and robustness, we can not ignore the security issues and pretend that we didn't notice. **Our work firstly prove that EEG BCI can be injected with invisible and robust backdoor, and unfortunately the backdoor can not be detected by any existing methods**. Our work provides valuable insight about the security issues of EEG BCI and promotes the draft of regulations for limiting Backdoor Attack in BCI.
>
> > Potential hazard in academia
>
> 5. Let's set apart the practical application of EEG BCI and talk about the academia. As we all know, EEG BCI is not only a medical devices, but also a good neuroimaging techniques because of its portability and low cost, which is widely used in the neuroscience society. Our attack can also be misused for **falsifying EEG datasets and drawing completely wrong neuroscience conclusion** as we discussed in the General Response. Our work promotes the construction of a more honest neuroscience society as we pointed out this cheating methods.
> **We think this cheating is probably the most severe hazard that can work now**.
>
> Best,
> Authors

---

### Author Response · Authors · 2024-11-19
**General Response by Authors**

We thank all the reviewers for your proficient and valuable comments and suggestions. We are cheerful to find that most of the reviewers have reached the consensus that our methods are novel and our idea of using reinforcement learning to optimize injection techniques is interesting (JuBu, uEuL, kEdv, dg1E). Moreover, we're also glad to see that our extensive experiments for showing the robustness (JuBu, uEuL, kEdv), the innovative three-stage attack (dg1E), the first multi-target attck on EEG BCI, effectiveness across multiple EEG tasks (uEuL, kEdv) are well recognized, and the research question is thought to be clear and easy to understand (JuBu).

**Here, we sincerely invite all the reviewers to read this general response before diving into the detailed responses to your individual concerns,** to help readers to understand **what we are contributing** and **the value of our work (or the severe potential hazard)**.

> What we are contributing.

Before the invention of Professor X, people may think EEG BCI is safe and cannot be injected with any backdoor, or the trigger in a backdoor poisoned sample is easily detected. However, our work alerts the EEG community that **EEG BCI is absolutely not safe**, and **EEG BCI can actually be injected with an invisible and robust backdoor attack**.

We conducted comprehensive experiments and validated the effectiveness of Professor X on various EEG tasks/formats. Moreover, we employed multiple backdoor defensive approaches to defend Professor X, but all of them failed. As a result, we didn't find an effective way to prevent Professor X.

Our work has determined that the **EEG BCI is not safe**, and **alert the whole community of the potential hazard** of misusing Professor X.

> The value of our work (or the severe potential hazard).

Some may worried that the application of EEG BCI is not broad, so regarding the safety of EEG BCI is lack of realism. However, **we don't have to wait until the horse has bolted before close the stable door**, that is, we don't need to wait for the widespread application of a technique before considering its security risk.

We admit that in clinical practice, the number of patients using EEG BCI is limited as they are all suffering severe illnesses like epilepsy [1] (for medical diagnosis) and paralysis [2] (for device control). However, **as long as the EEG BCI is not safe, the result of malicious manipulation is catastrophic for any single patient** -- like misleading the localization of epilepsy lesions, or controlling a paralytic to act dangerously.

**As long as there is one person may be harmed by the security risk of EEG BCI, studying the safety is meaningful.** Not to mention that there are **50 million** epilepsy patients and **15 million** patients suffering from spinal cord injuries worldwide (data from World Health Organization' website). Although only a small number of them have the conditions to receive BCI treatment, this is still not a small number. Since the EEG BCI already helps a paralytic walk again [2], the wider application of EEG BCI is predictable.

Besides the real-world security risk, our attack can also be misused for **falsifying EEG datasets**, which is **more severe in academia**. With the help of Professor X, one can essentially adjust the classification accuracy of a EEG dataset to any level (by adjusting the poisoning rate $\rho$). This will be a disaster for EEG research community, since one can **draw any neuroscience conclusion from a fake EEG dataset** by misuing Professor X.

For example, Jack is an EEG researcher and he wants to know whether can we detect Alzheimer's disease from EEG. So he put in great effort to build an large dataset recording 100 older's EEG signals (half are Alzheimer patients). But he runs classification model on this dataset and finds that the accuracy is very low, let's say just 52% (random level is 50%). Jack doesn't want to see his efforts go to waste, and **unethically used Professor X** to falsify his dataset, increasing the accuracy to 80%. Finally, Jack claims he find that he can detect Alzheimer's disease from EEG signals. However, this neuroscience conclusion is totally a fake finding.

**Every coin has two sides**, our attack can be adopted for protecting intellectual properties of EEG datasets and BCI models. But most importantly, we want to **call for defensive research** as we have not found an effective way to detect and defend Professor X.

[1] Shoeb A, Guttag J. Application of machine learning to epileptic seizure detection[C]//Proceedings of the 27th International Conference on International Conference on Machine Learning. 2010: 975-982.

[2] Lorach H, Galvez A, Spagnolo V, et al. Walking naturally after spinal cord injury using a brain–spine interface[J]. Nature, 2023, 618(7963): 126-133.

---

### Comment · Area_Chair_r9hc · 2024-11-21
**Rebuttal**

Dear Reviewers,

I encourage you to review the rebuttal and reach out to the authors with any additional questions or requests for clarification.

Best,\
AC

---

### Author Response · Authors · 2024-11-23
**Kindly request for post-rebuttal comments**

Dear Reviewers:
﻿

Thank you again for your wisdom and valuable comments. We have provided additional results or complete explanations for all the questions. Since the rebuttal process is approaching its end, we would be glad to hear from you whether our rebuttal has addressed your concerns. Feel free to comment on our rebuttal if you have further questions or considerations.

---

### Comment · Area_Chair_r9hc · 2024-11-25

Dear Reviewers,

As we approach the end of the discussion period, I would like to encourage you again to review the rebuttal and engage with the authors if you have any additional questions or need further clarification. If you have no further questions, please acknowledge that you have received and reviewed the rebuttal.

Best,\
AC

---

### Author Response · Authors · 2024-11-26
**Adding a discussion section about defensive study in the revised paper**

Dear ACs, reviewers, and readers:

Thanks to the reviewer JuBu, uEuL and kEdv, who asked many questions regarding the defensive study against Professor X. These insightful concerns deepen our understanding of our attack and how to guard backdoor attack in EEG BCIs. Thus, we added a new section in the revised paper's appendix (marked in blue) to discuss our humble opinion on the defensive study against Professor X, which we hope will benefit the future research.

Best,

Authors

---

### Comment · Area_Chair_r9hc · 2024-11-29

Dear Reviewers,

If you haven’t already done so, I strongly encourage you to engage with the authors. Please note that there is no need to make any commitments regarding the final score at this time; but I would appreciate it if you could acknowledge that you have received and reviewed their responses, and ask any follow-up questions you may have.

Best,\
AC

---

### Meta-Review · Area_Chair_r9hc · 2024-12-19

**Metareview:**

The paper introduces a novel and robust backdoor attack framework for EEG-based Brain-Computer Interfaces. The method employs a label-poisoning strategy with reinforcement learning to inject undetectable triggers into EEG signals. The paper introduces and tackles a very interesting notion - one which should not be ignored as BCI technologies become more adopted. The solution is also interesting and effective. However, the paper had a few weaknesses. The paper is somewhat difficult to follow, with key concepts like "stealth" lacking clear definition and justification. It also falls short in benchmarking against recent methods in time-series or signal-processing attacks. Additionally, the absence of an ethical discussion is a significant oversight for a topic of this nature. The analysis of backdoor defense mechanisms is limited in both scope and depth.

**Additional Comments On Reviewer Discussion:**

The paper received highly diverging reviews of 1, 5, 6, 6, 8. Here is a summary of the discussions and the rationale for my decision:

- Reviewer RejD who gave a **score of 1**, explicitly mentioned that "The notion of a  BCI backdoor vulnerability is unfounded". I respectfully disagree with this and applaud the authors for exploring a novel, interesting, and forward-thinking problem in the sensitive area of BCI.

- However, other issues were also identified. The paper was found to be somewhat difficult to understand. For instance, the concept of "stealth" felt unclear and lacked proper definition and justification. Another shortcoming is the absence of benchmarking against recent time-series or signal-processing attack methods. Additionally, the paper does not include a clear discussion of ethics, which is essential for a topic like this. The analysis was generally found to be a bit limited in scope and depth w.r.t. backdoor defense mechanisms.

- Reviewer RejD agrees with these shortcomings.

- Reviewer uEuL who gave a **score of 8** *agrees* with the shortcomings regarding clarity and insufficient benchmarking, *however, strangely did not change their score after agreeing with these key issues*.

- Reviewer dg1E also agrees with the limitations of the benchmarking and analysis in the paper.

Overall, taking everything into account and reducing the impact of the two extreme scores (1 and 8) due to the reasons mentioned above, I believe the paper is still slightly below the threshold for acceptance. However, I highly encourage the authors to continue this line of work and further polish the paper for another round of submission. Better presentation and clarity of the work (e.g., specific definitions), followed by more benchmarking and analysis/discussions, will make the paper acceptable in my opinion.

---

### Decision · Program_Chairs · 2025-01-22

Reject